# NOISE RE-SAMPLING FOR HIGH-FIDELITY IMAGE GENERATION

## ABSTRACT

Latent diffusion models (LDMs) have emerged as powerful tools for generating diverse and realistic samples across domains. However, their efficacy in capturing intricate details and small-scale objects remains a challenge. Our investigation reveals that VAE compression induces errors in the latent space and limits the generation quality. Furthermore, LDMs trained on fixed-resolution images struggle to produce high-resolution outputs without distortions, making simple resolution increases ineffective. In this paper, we propose a novel **noise re-sampling** strategy that enables multi-scale generation of LDMs, allowing LDMs to "zoom in" and improve generation quality of local regions. By increasing the sampling rates from the noise perspective in the latent space, we effectively bypass the constraints imposed by VAE compression, thus preserving crucial high-frequency information. Our approach, a simple yet effective plugin for current LDMs, enhances the quality of image generation in local regions while maintaining overall structural consistency and providing fine-grained control over the scale of generation in latent diffusion models. Through extensive experimentation and evaluation, we demonstrate the efficacy of our method in enhancing the generation quality across various LDM architectures. Our approach surpasses existing methods, including stable diffusion (SD) models, SD-based super-resolution methods and high-resolution adaptation methods, in generating high-fidelity samples of complex objects.

## 1 INTRODUCTION

In recent years, the advance of latent diffusion models (LDMs) (Ho et al., 2020b; Rombach et al., 2022; Diffusion, 2022; Podell et al., 2023; Wu et al., 2023; Singer et al., 2022) has marked a significant leap in the field of generative modeling, offering a new paradigm for creating diverse and lifelike samples across various domains. These models utilize a latent space, compressed by Variational Autoencoders (VAEs) (Kingma & Welling, 2013), to efficiently generate detailed and coherent images. Despite their considerable capabilities, LDMs frequently encounter challenges in accurately rendering complex objects such as hands, faces, and textual elements (Lu et al., 2023; Guo et al., 2025). This deficiency primarily stems from the compression errors associated with the latent space where VAEs tend to disproportionately affect high-frequency details—details that are crucial for the realism and perceptual quality of the generated samples. While intuitively, increasing the resolution of LDMs with higher sampling rates might seem like a solution, LDMs trained on fixed-resolution images (Diffusion, 2022; Podell et al., 2023; Chai et al., 2022) struggle to generate high-resolution outputs without introducing distortions (Zheng et al., 2023; He et al., 2024; Guo et al., 2025), making simple resolution increases ineffective.

Efforts have been made to enhance LDMs for generating high-resolution content to improve visual details (Podell et al., 2023; Guo et al., 2025; Zheng et al., 2023; He et al., 2024; Jin et al., 2023; Ho et al., 2022). These methods, which include both fine-tuning (Podell et al., 2023; Zheng et al., 2023; Ho et al., 2022; Guo et al., 2025) and tuning-free (He et al., 2024; Jin et al., 2023; Si et al., 2023) approaches, adapt LDMs trained at a fixed resolution to operate at higher or lower resolutions. However, these methods often necessitate precise adjustments of parameters like the dilated stride (He et al., 2024) and additional steps (Guo et al., 2025). Such calibration is necessary, without which potentially leads to inconsistencies in the quality of the generated images. Fine-tuning or adaptor-based (Guo et al., 2025; Hu et al., 2021; Wang et al., 2023) methods requires extensive high quality high resolution data and incurs significant computational expenses due to exponentially increased

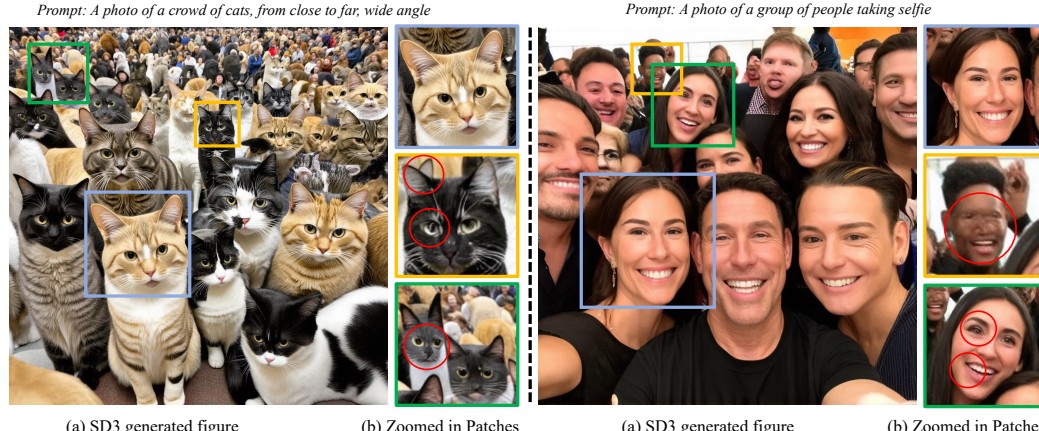

*Prompt: A photo of a crowd of cats, from close to far, wide angle*                    *Prompt: A photo of a group of people taking selfie*

(a) SD3 generated figure          (b) Zoomed in Patches          (a) SD3 generated figure          (b) Zoomed in Patches

Figure 1: Examples of text-to-image generation results using SD 3 (Esser et al., 2024a). Zoomed-in patches are provided for clearer examination of details. When generating objects of the same category (e.g., cats, human faces), the scale of the objects significantly influences generation quality. Large-scale regions (highlighted in blue) exhibit rich details and accurate structures, while small-scale regions (highlighted in yellow and green) suffer from poor detail and structural inaccuracies (marked by red circles).

input size. Nevertheless, both tuning-free and fine-tuning high-resolution adaptation methods necessitate LDMs to produce images at resolutions not included in the pre-training, thereby limiting the generative capability of LDMs and potentially leading to inconsistent output distribution to original LDMs. Moreover, these methods inherently suffers from the inability of LDMs at reproducing intricate details or small-scale objects Guo et al. (2025).

In this paper, we focus on improving the generation quality of LDMs at complex details from a frequency domain perspective. Our findings reveal that the quality of generated objects in LDMs is strongly influenced by their size relative to the overall image resolution, i.e. the scale of objects. Large-scale objects within the image benefit from more detailed reconstruction, while small-scale objects, confined to limited regions, are often poorly reproduced, as demonstrated in Figure 1. Additionally, we observe that most artifacts associated with small-scale objects or complex details are localized in the high-frequency band (Lin et al., 2023; Kingma et al., 2019). The use of Variational Autoencoders (VAEs), commonly employed in LDMs to compress data into a more manageable latent space, exacerbates this issue by diminishing fidelity in these high-frequency components. These components are crucial for capturing fine textures and subtle contours and are particularly important for small-scale objects. As a result, the loss of high-frequency details leads to images that are overall structurally accurate yet lacking sharpness and textural nuances or with noticeable artifacts on small-scale objects. Moreover, because LDMs operate exclusively in the latent space, mitigating this degradation remains challenging, even with the availability of higher-quality training data for the denoising model. Consequently, the generation quality of LDMs is directly impacted.

To address the above issues, we propose an novel **Noise Re-sampling** approach to enable the multi-scale generation ability of LDMs. By adjusting sampling rates at local regions, that are with complex details or contains small-scale objects, from the noise perspective, our method improves generation quality and retores high-frequency details in these regions, harnessing the intrinsic generative capacity of LDMs at their native resolution. As a simple yet effective plugin to current LDMs, this approach allow us to bypass the constraints widely imposed by VAE compression and restore high-frequency information pivotal in achieving high-fidelity image synthesis. Furthermore, to maintain consistency between the re-sampled local regions and the original image while increasing the effective sampling rate, we introduce a VAE-based upscaling method. This method upscales the latents to serve as an accurate guided for the re-sampling process, avoiding the issues associated with flawed direct latent-space interpolation.

Extension experiments are conducted for comprehensive comparison with existing methods, including stable diffusion models, SD-based super-resolution methods and high-resolution adaptation

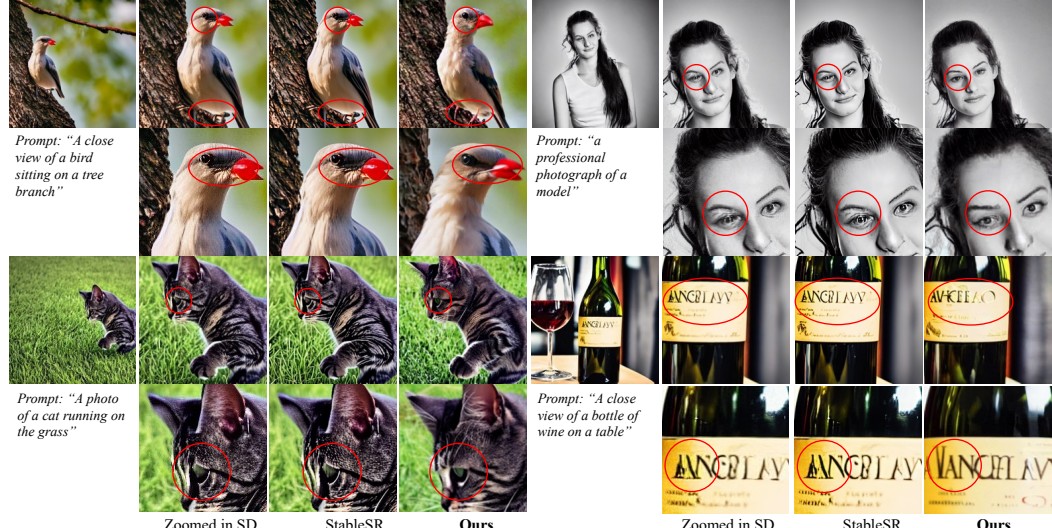

Figure 2: Examples of text-to-image generation results of SD (Diffusion, 2022), StableSR (Wang et al., 2023) and proposed Noise Re-sampling approach. Zooming in for better detail. For each set of images, the top-left most image is the original SD generated image, which contains noticeable artifacts. StableSR, while enhances the sharpness of local patches, it struggles to recover intricate details (highlighted in red). In comparison, our method not only improves the details of local regions but also effectively corrects distortions and artifacts.

approaches. Through rigorous experimentation and qualitative assessments, we demonstrate the superiority of our method in producing samples that closely mimic the complexity and detail of their real-world counterparts.

## 2 RELATED WORK

**Latent diffusion models**. Latent Diffusion Models (LDMs) (Rombach et al., 2022; Diffusion, 2022; Podell et al., 2023) represent a significant advancement in the field of generative models, particularly in the domain of image generation. Introduced as an efficient alternative to traditional diffusion models, LDMs operate in a compressed latent space rather than the pixel space, drastically reducing computational demands and enabling faster model training and sampling. This approach leverages an Variantional Autoencoder (VAE) (Kingma & Welling, 2013; Oord et al., 2017) to map high-dimensional data into a lower-dimensional latent space. The diffusion process is then applied within this latent space, preserving the ability to generate high-quality images while enhancing the model's efficiency. LDMs have demonstrated remarkable performance in tasks such as image synthesis, super-resolution, and conditional image generation, establishing them as a pivotal development in generative deep learning.

**Any resolution training**. While existing latent diffusion methods (Diffusion, 2022; Podell et al., 2023) excel in image synthesis, they struggle to generate high-resolution images due to the complexities of high-dimensional data and the scarcity of high-quality datasets. The primary approaches (Podell et al., 2023; Ho et al., 2022; Teng et al., 2023) include cascaded models that initially produce low-resolution images and then incrementally upscale them to higher resolutions. Alternatively, end-to-end models (Podell et al., 2023; Hoogeboom et al., 2023) directly create high-resolution images but require extensive training and substantial datasets. Other strategies involve fine-tuning (Hu et al., 2021), where only part of the model's parameters are adjusted to adapt to higher resolutions, necessitating numerous tuning steps. Additionally, some recent methods (He et al., 2024; Si et al., 2023) propose training-free approaches, employing techniques like dilated convolution to adjust convolutional networks for different resolutions. However, these methods can introduce semantic inconsistencies and visual artifacts.

**Super-Resolution** Super-resolution techniques employ generative models to create high-resolution images by exploiting the self-similarity attributes of images, refining the output from these models (Wang et al., 2023; Sun et al., 2023; Yue et al., 2024). These techniques generally tackle a localized, conditional challenge, relying heavily on the structure provided by the initial low-resolution input. Consequently, when applied to synthetic data from Latent Diffusion Models (LDMs), which may contain distorted structural information, super-resolution methods might struggle to correct these errors effectively. Our method takes a different approach by starting in the noise space and applying a re-sampling method to increase sampling rates in complex local regions. This technique helps recover the high-frequency details and correct distorted structural information that were originally lost, offering a robust solution to the limitations of traditional super-resolution methods.

## 3 PRELIMINARY

Diffusion models (Ho et al., 2020a; Rombach et al., 2022; Diffusion, 2022; Podell et al., 2023) are generative models that simulate a process where data gradually transitions into Gaussian noise and then learns to reverse this process to generate new data. This method has shown exceptional capabilities in generating high-quality images, audio, and text. In DDPM (Ho et al., 2020a), the forward process transforms data $x$ into noise through a series of steps, each adding Gaussian noise:

$$q(x_{t+1}|x_t) = \mathcal{N}(x_{t+1}; \sqrt{1-\beta_t}x_t, \beta_t I) \tag{1}$$

where $\beta_t$ are predetermined noise levels. The reverse process then attempts to reconstruct the original data from this noise, optimized via a denoising network $\mu_\theta$ parameterized by $\theta$:

$$p_\theta(x_{t-1}|x_t) = \mathcal{N}(x_{t-1}; \mu_\theta(x_t, t), \sigma_t^2 I) \tag{2}$$

VAEs (Kingma et al., 2019) compress data $x$ into a latent representation $z$ and then reconstruct $x$ from $z$. The encoder parameterized by $\phi_E$ learns the distribution parameters:

$$q_{\phi_E}(z|x) = \mathcal{N}(z; \mu_{\phi_E}(x), \sigma_{\phi_E}(x)^2 I) \tag{3}$$

The decoder, parameterized by $\phi_D$ rebuilds $x$ from $z$ aiming to minimize the reconstruction error and the KL divergence between the learned latent distribution and the prior:

$$p_{\phi_D}(x|z) = \mathcal{N}(x; \mu_{\phi_D}(z), \sigma_{\phi_D}(z)^2 I) \tag{4}$$

Together, VAEs and Diffusion models offer an efficient framework for generating detailed, realistic samples by effectively managing the latent space and refining the generation process.

## 4 LATENT SPACE NOISE RE-SAMPLING

### 4.1 PROBLEM FORMULATION

Latent Diffusion Models (LDMs) (Ho et al., 2020b; Diffusion, 2022; Rombach et al., 2022) denoise images by gradually refining noise samples drawn from a Gaussian distribution. However, one crucial aspect, the sampling rate of the noise samples, has been largely overlooked, which directly decides how many pixels are allocated to an object and affects the quality of the generated object in image. In this work, we conceptualize each image as a discrete sampling from a bounded, continuous scene, structured within a normalized coordinate domain of $[0, 1]^2$. Under this formulation, current LDMs (Ho et al., 2020b; Diffusion, 2022; Podell et al., 2023) typically produces discrete images at a limited sampling rate $1/\Delta$, where $\Delta$ denotes the spatial distance between pixels. The introduction of VAEs, which compress image data into compact latent codes for efficiency and scalability, further reduces the sampling rate, leading to distorted and low-quality results, as shown in Figure 3.

To address these problems, we propose a novel **Noise Re-sampling** approach, which adjusts sampling rate at local regions with complex details or small-scale objects and bypass the negative impact imposed by VAE compression. The overall pipeline begins with the generation of a global image using the standard LDM denoising process as shown in Figure 4 (b). Local regions requiring enhancement are identified, and their noise patches are re-sampled from the same noise space at an increased sampling rate to restore fine-grained details. These high-resolution noise patches are then denoised, decoded into image space and seamlessly merged back into the global image. To ensure

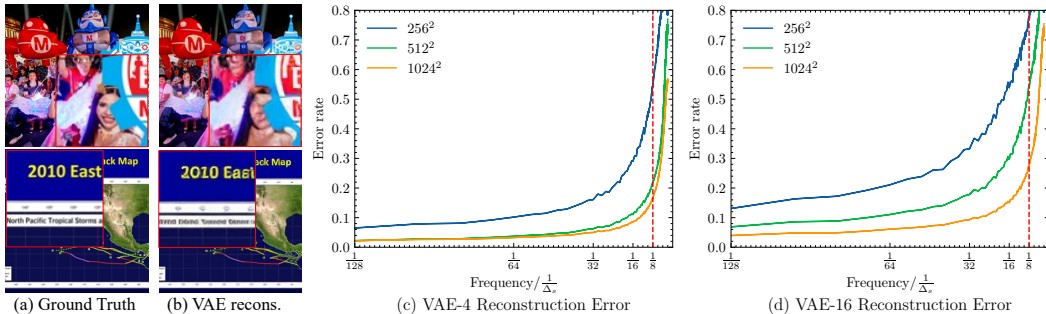

(a) Ground Truth    (b) VAE recons.    (c) VAE-4 Reconstruction Error    (d) VAE-16 Reconstruction Error

Figure 3: Examples of ground truth and VAE reconstructed image from Laion5B (Schuhmann et al., 2022) (left). Frequency domain analysis of VAE reconstruction error rates at different resolutions on Laion5B (right) with (c) 4 channel (VAE-4) and (d) 16 channel latent space (VAE-16), illustrating the degradation of high-frequency components in VAE reconstruction, despite the channel size. In addition, with increased resolution (i.e., sampling rate), the overall recontruction error is substantially reduced. For example, for frequency components with $f = 1/8\Delta_s$, the reconstruction error for input of resolution $1024^2$ is much lower than $512^2$ and $256^2$.

consistency and minimize artifacts, upsampling is performed in the image space using a VAE-based process before being compressed back into the latent space. Finally, we can obtained an image with enhanced local details while maintaining global coherence

In Sec. 4.2, we analyze VAEs from a frequency perspective and demonstrate their adverse effect on the generation quality of LDMs. In Sec. 4.3, we reformulate the LDMs pipeline from a signal processing viewpoint and introduce our Noise Re-sampling approach. In Sec. 4.4, we present a VAE-augmented upscaling technique that shifts the upsampling operation from the latent space to the image space, preventing artifacts and blurring in the output and providing an accurate guide for Noise Re-sampling.

## 4.2 VAEs are Lossy

In current LDMs, Variational Auto-Encoders (VAEs) are commonly integrated for efficiency, compressing image data into compact latent representations. However, this compression is inherently lossy, particularly affecting high-frequency components critical for fine details, as shown in Figure 3. The decoder attempts to reconstruct the original image from the latent representation but cannot fully recover these lost details, leading to diminished generation quality. Since LDMs operate solely in the latent space, addressing this issue is challenging, even with high-quality training data for the denoising model, making it a significant bottleneck for achieving high-fidelity results.

**Quantifying Error in VAEs**. To quantify the error of the VAE reconstruction process, we use real-world data $x_n$ and feed them into the VAE encoder-decoder pipeline to obtained the reconstructed sample $\hat{x}_n$, and measure the error in the frequency domain. The error at frequency $f_i$ is denoted as:

$$\mathcal{E}(f = f_i) = \sum_{n=1}^{N} ||\mathcal{F}_{f=f_i}(x_n) - \mathcal{F}_{f=f_i}(\hat{x}_n)||_2^2 \qquad (5)$$

where $\mathcal{F}$ represents the Discrete Fourier Transform (DFT) (Gao et al., 2021; Lin et al., 2023) and $N$ is the total number of samples.

As shown in the Figure 3 (c) and (d), when analyzing the reconstructed images in the frequency domain, it becomes evident that while the decoder attempts to recover much of the lost information, it struggles significantly with the high-frequency components (the intricate details), leading to distorted faces and texts (Figure 3 (b)). Consistent to our analysis, the components with higher frequencies are likely to have larger error (Figure 3). The loss of high-frequency details results in images that appear distorted and corrupted than the original scene, highlighting a fundamental limitation in the current VAE architecture. Furthermore, as the resolution increases, the overall error in the high-frequency bands decreases significantly, demonstrating the effectiveness of higher sam-

pling rates. This observation motivates us to propose the noise re-sampling approach to bypass the constraint of VAE compression.

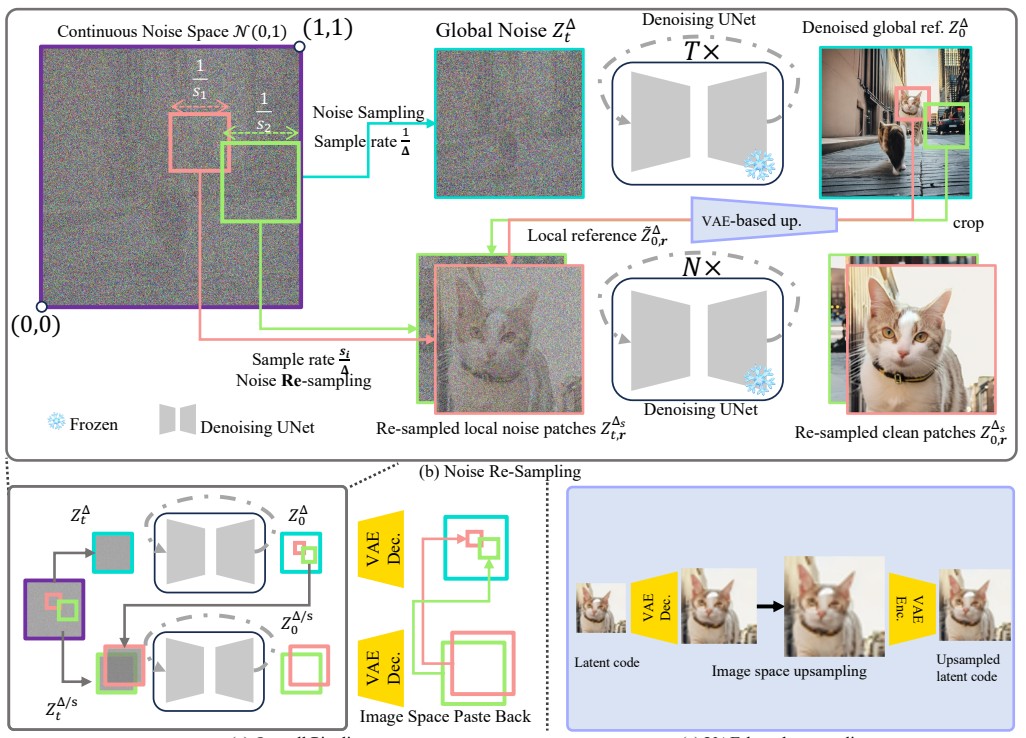

Figure 4: Illustration of proposed Noise Re-sampling Method. The overall pipeline is demonstrated in (a). Noise are initially generated within a bounded continuous scene, with normalized coordinate of $[0,1]^2$. Global noise $Z_t^\Delta$ is first sampled from continuous noise with sample rate $1/\Delta$ and denoised to serve as reference. (b) The proposed Noise Re-sampling samples local patches $\mathbf{r}$ of high-sampling rate noise $Z_{t,\mathbf{r}}^{\Delta_s}$, combined with local reference cropped from $Z_{0,\mathbf{r}}^{\Delta_s}$ to generate detailed sub-regions without altering the global image resolution. (c) VAE-based upsampling upsamples local reference patches in the image space to maintain the fidelity while matching the resolution of re-sampled noise, avoiding flawed latent-space upsampling.

### 4.3 NOISE RE-SAMPLING

In LDMs, the quality of generated objects is heavily influenced by their size relative to the image (Zheng et al., 2023). Larger objects benefit from more allocated pixels, enabling more detailed reconstruction, whereas smaller objects suffer from reduced resolution. Such phenomenon coins with the concept of **sampling rate**. The sampling rate, typically defined as the number of pixels per unit area, quantifies how well an object or scene is represented in an image. According to the Nyquist-Shannon sampling theorem, high-frequency details are inherently limited by the sampling rate. Thus, when the sampling rate is low—meaning fewer pixels are allocated to an object—capturing its finer details becomes increasingly difficult. Compounding this issue, current LDMs operate extensively within the compressed latent space of VAEs to enhance computational efficiency, which further reduces the effective sampling rate.

Given the strong correlation between the reconstruction error in VAEs and the frequency of components, our objective is to minimize this error through a Re-sampling approach by increasing sampling rate at complex local regions. Specifically, since LDMs denoise noise samples to generate images without altering their resolution, the sampling rate of the resulting image is directly determined by the sampling rate of the noise sample. Hence, our re-sampling approach begins within the noise space itself.

**Noise Space from Signal Processing View.** Assume $Z_t(x, y)$ to be an infinite, continuous 2-D signal in the noise space at timestep $t$, as shown in Figure 4 (b), which, after denoising, represents a real continuous scene. However, in practice, we can not directly work on continuous signals. Therefore, current LDMs often starts with a bounded, discrete noise sample $Z_t^\Delta$ with sampling rate $1/\Delta$. We reformulate the process of initial noise sampling in the noise space as a bounded discretization process of $Z_t(x, y)$:

$$Z_t^\Delta[n, m] = Z_t(n\Delta, m\Delta), n \in [0, H], m \in [0, W] \tag{6}$$

Under such formulation, complex objects (e.g., hands, faces and etc.) are denoised with same sampling rate as other regions, such as background. As a result, these objects are poorly reproduced and has distorted details. To faithfully reconstructs complex regions, we can increase the sampling rate of them during the denoising process and obtain final clean latent $Z_0^\Delta$ with better detail. Consequently, the high frequency component associated with these intricate details can be recovered. However, when the resolution exceeds the training resolution of LDMs, the results are commonly observed with issues such as pattern repetition, distorted object structure (Si et al., 2023; He et al., 2024), causing simple increase in sampling rate ineffective. To tackle this problem, we propose the noise re-sampling approach that adaptively increases sampling rates of local regions without exceeding the native training resolution of LDMs, thereby fully utilizing the generative power of LDMs and enhancing generation quality.

**Noise Re-Sampling.** As shown in Figure 4 (b), To fully exploit the generation ability of LDMs at their native resolution while enabling multi-scale generation ability, we increase sampling rate at local regions. Specifically, we re-sample noise from the continuous noise space for the local region $\mathbf{r}$ with size $1/s$ in normalized coordinates $[0, 1]^2$, with bottom-left point $(l_x, l_y)$. :

$$Z_{t,\mathbf{r}}^{\Delta_s}[n, m] = Z_t^{\Delta_s}(n\frac{\Delta}{s} + l_x, m\frac{\Delta}{s} + l_y), n \in [0, H], m \in [0, W] \tag{7}$$

where $\Delta_s = \Delta/s$ denotes the increased sampling rate at local region $\mathbf{r}$. However, directly denoising the patch $Z_t^{\Delta_s}$ may lead to inconsistencies between local and global images, primarily due to the inability of LDMs to "zoom in" (Zhang et al., 2023b).

**Local Reference.** To address this challenge and maintain consistency between re-sampled patches and global reference without extensive re-training, we propose an alternative approach. We first crop the corresponding local region $\mathbf{r}$ from $Z_0^\Delta$ and upscale it as $Z_{0,\mathbf{r}}^\Delta$ to serve as a reference before denoising $Z_{t,\mathbf{r}}^{\Delta_s}$. Specifically, we diffuse the cropped region back into the noise space (the forward process of DDPM) (Ho et al., 2020a) using the following equation:

$$\tilde{Z}_{t,\mathbf{r}}^{\Delta_s} = \sqrt{\alpha_t}\mathcal{U}(Z_{0,\mathbf{r}}^\Delta) + \sqrt{1 - \alpha_t}\epsilon, \epsilon = Z_{t,\mathbf{r}}^{\Delta_s} \in \mathcal{N}(0, 1) \tag{8}$$

where, $\mathcal{U}$ denotes the upsampling operation to match the resolution between re-sampled noise $Z_{t,\mathbf{r}}^{\Delta_s}$ and local reference $Z_{0,\mathbf{r}}^\Delta$. To further ensure consistency (Mokady et al., 2023), $t$ is set to an intermediate step $N$.

The final clean local re-sampled latent is obtained by denoising $\tilde{Z}_{t,\mathbf{r}}^{\Delta_s}$ to $\tilde{Z}_{0,\mathbf{r}}^{\Delta_s}$ using Eq. 2. As a result, the sampling rate is effectively increases by a factor of $s$ within the selected local regions. The high frequency components can be accurately generated within this local area. This enhancement allows for the preservation of more detailed information and faithful reconstruction of small-scale objects. Moreover, with this more detailed and accurate information available, the VAE decoder is better equipped to decode the high-sample-rate clean latent code with improved precision and quality. The final re-sampled clean latent code can be directly paste back to the corresponding local region of the original output after decoding.

Overall, this approach allows us to sample noise at a finer granularity, thereby expanding the effective frequency spectrum that our latent diffusion models can utilize. By enhancing the sampling rate at which noise is sampled, our models are better equipped to reconstruct the intricate details that contribute to the visual fidelity of the generated images.

## 4.4 VAE-BASED UPSCALING

As previously noted, when re-sampling a local region in proposed Noise Re-sampling approach, it is essential to have a corresponding crop in the low-quality global image as a reference. This

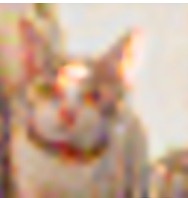 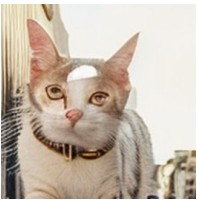 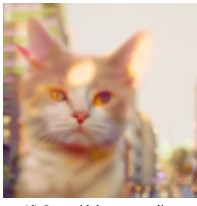 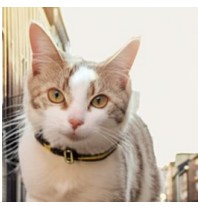

(a) Original latent     (b) Latent space upscaling     (c) VAE-based upscaling     (d) Ours with latent upscaling     (e) Ours with VAE-based upscaling

Figure 5: Visualization of reconstructed images from latent codes under different upsampling methods. (a) original latent code. (b) results of direct latent space upsampling (decoded to image space), which has distorted and blurry semantic content. (c) the VAE-based upsampling process, preserving better detail and semantic consistency to original latent. (d) illustrates noise re-sampling using direct latent space upsampled latent code as reference, resulting in distortion and deviation from the original reference. (e) highlights our method with VAE-based upsampling, which significantly improves image quality and consistency for both the upsampled reference latent and final re-sampled clean latent.

reference ensures that the denoising process aligns with the overall image, preventing deviations that could lead to inconsistencies across the global image. However, it's important to note that direct upsampling in the latent space does not function equivalently to upsampling operations in the image space. In the latent space, upsampling can distort the semantic content encoded within the latent vectors, leading to artifacts in the resulting images as shown in Figure 5 (b). To mitigate this issue, we propose a VAE-based upsampling method where the latent vectors are decoded back to the image space for upsampling.

As shown in Figure 4 (c), before upscaling, we employ the Variational Autoencoder (VAE) decoder to project latent vectors back into image space. The VAE decoder reconstructs compressed latent patch to image space, including its high-frequency components. This reconstruction is not a straightforward interpolation rather, it generates a new, higher-resolution image with reconstructed high frequency components based on its learned distribution of image features. With latent codes decoded into the image space, we upsample the decoded image by $s$ to match the increased sampling rate at local region in proposed Noise Re-sampling, demonstrated in Figure 5 (c).

Once the VAE has provided this initial, imperfect high-resolution output, we employ the proposed Noise Re-sampling process. This subsequent phase is tailored to specifically target and refine the inaccuracies in the high-frequency details originally presented in the low-sampling-rate latent code as well as potential artifacts introduced by the VAE, as shown in Figure 5 (e). Through noise re-sampling, this process iteratively adjusts and sharpens these details, enhancing the overall fidelity of the image while preserving structural consistency.

## 5 EXPERIMENTS

### 5.1 IMPLEMENTATION DETAILS

The proposed method is implemented using PyTorch and evaluated on 8 NVIDIA V100 GPUs. We utilize the base models SD 1.5, SD 2.0 (Diffusion, 2022) and SD 3 (Esser et al., 2024a), with all parameters frozen. The inference resolutions are $512^2$, $768^2$ and $1024^2$ for SD 1.5, SD 2.0, SD 3, respectively. Experiments on human-centric image generation adopt ControlNet (Zhang et al., 2023a) with DWpose (Yang et al., 2023) estimated poses. Across experiments, we adopt the DDIM (Song et al., 2020) sampling scheduler with 50 steps. $N$ is set to be half of the total denoising steps to balance between quality and consistency. For text-to-image tasks, the positions of local regions are randomly selected and kept consistent across methods. As for human-centric image generation, the local regions are centered at coordinates of hands or faces based on DWpose extracted points. We evaluated two scenarios: global image refinement and local region generation quality. The global scenario assessed how our method enhances overall quality by refining local regions with metrics $FID_g$ and $KID_g$ following (He et al., 2024; Parmar et al., 2022). The local scenario focused solely on the quality of local patches, measuring the model's ability to "zoom in" effectively with $FID_l$ and $KID_l$ (He et al., 2024; Parmar et al., 2022). Specifically, we take random patches from original LDM

generated images and apply each method to enhance the local image quality. In addition to FID and KID, we adopt CMMD (Jayasumana et al., 2024) to measure the re-sampled local regions, which offers a more robust and reliable assessment of image quality of current text-to-image models.

## 5.2 EVALUATION ON TEXT PROMPT IMAGE GENERATION

Table 1: Evaluation results on SD 1.5 and SD 2.0 (Rombach et al., 2022; Diffusion, 2022). Comparison of overall and local image quality across methods measured with FID ↓, KID ↓ (Parmar et al., 2022) and CMMD ↓ (Jayasumana et al., 2024). Our method demonstrates competitive performance, particularly in enhancing local details.

| Model | SD 1.5 | | | | | SD 2.0 | | | | |
|---|---|---|---|---|---|---|---|---|---|---|
| Method | $FID_g$ | $FID_l$ | $KID_g$ | $KID_l$ | CMMD | $FID_g$ | $FID_l$ | $KID_g$ | $KID_l$ | CMMD |
| SD | 10.59 | 15.60 | 0.0019 | 0.0044 | 0.640 | 9.66 | 14.19 | 0.0020 | 0.0033 | 0.410 |
| Stable SR | 11.56 | 16.70 | 0.0034 | 0.0052 | 0.613 | 15.75 | 17.31 | 0.0063 | 0.0086 | 0.680 |
| MultiDiff | 36.68 | 35.54 | 0.0143 | 0.0127 | 0.746 | 32.42 | 31.35 | 0.0140 | 0.0121 | 0.607 |
| ScaleCrafter | 17.42 | 21.26 | 0.0035 | 0.0048 | 0.509 | 18.73 | 23.19 | 0.0087 | 0.0049 | 0.474 |
| Ours | **10.40** | **14.03** | **0.0018** | **0.0035** | **0.464** | **9.56** | **13.94** | **0.0019** | **0.0032** | **0.293** |

**Dataset and Evaluation.** Laion-5B (Schuhmann et al., 2022) is adopted for evaluation of text-to-image generation which contains 5 billion images with captions. 20k images are randomly selected along with captions as text prompt.

**Comparison to super-resolution method.** We compared our method to StableSR, a super-resolution baseline by Wang et al. (2023) (Wang et al., 2023). While StableSR improves sharpness and reduces aliasing, it fails to correct SD-induced distortions (Figure 2). StableSR's fine-tuning with high-resolution data leads to results deviating from the original SD distribution with limited performance when evaluated against training data of SD. In contrast, our training-free approach is adaptable to various LDMs with minimal cost. Generating at native resolution, our method ensures consistency with the original SD distribution. As shown in Table 1, our method surpasses StableSR in enhancing local patch quality while maintaining overall consistency.

**Comparison to high-resolution adaptation method.** Our method differs from ScaleCrafter (He et al., 2024), which heavily relies on hyper-parameter adjustments for higher resolutions, often requiring careful tuning to avoid artifacts. Generated at non-native resolution of LDMs, the results potentially deviates from the original SD output distribution. In contrast, our method utilizes noise re-sampling directly in the latent space at native resolution of LDMs. In addition, compared to patch-based method MultiDiff Bar-Tal et al. (2023), our approach exhibits superior ability at enhancing local details, whereas MultiDiff merges patches without improving patch-wise generation quality. Experiment results in Table 1 show our method outperforms ScaleCrafter and MultiDiff in detail enhancement and overall image quality.

## 5.3 EVALUATION ON HUMAN-CENTRIC IMAGE GENERATION

**Dataset and Evaluation** The UBC-Fashion validation dataset is employed for evaluations in human-centric image generation tasks. This dataset consists of 38,969 video frames featuring human models in various poses. Using DWpose (Yang et al., 2023), we extract pose images from these frames to serve as conditions for ControlNet (Zhang et al., 2023a). Text prompts are selected from the Laion-Human datasets (Ju et al., 2023a;b), providing a contextual basis.

**Comparison to baseline.** Our approach was compared to SD baseline with ControlNet (Zhang et al., 2023a), focusing primarily on the generation of detailed and realistic images of human subjects, particularly in challenging areas such as faces and hands. The results shown in Table 2 confirmed that our method could maintain high fidelity in detail while enhancing specific regions of interest, such as faces and hands, without compromising the overall structural integrity of the images. When evaluated against Laion-5B samples, our approach cannot match the performance of baseline primarily due to parsity of human-centric images in Laion-5B samples. Performance on UBC-fashion justifies our analysis.

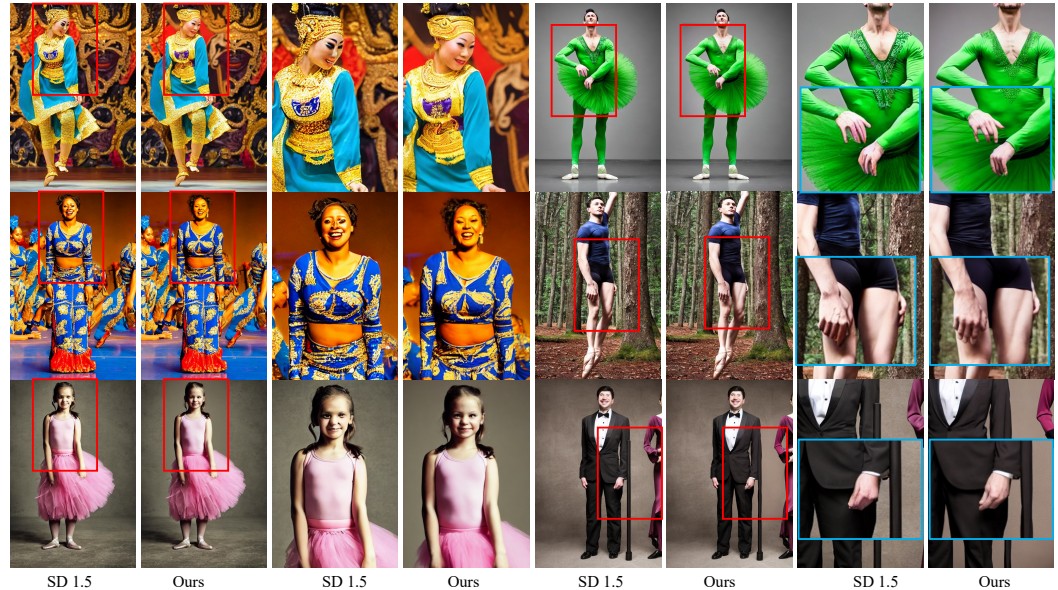

SD 1.5    Ours    SD 1.5    Ours    SD 1.5    Ours    SD 1.5    Ours

Figure 6: Visual quality comparisons between SD 1.5 + Controlnet (Zhang et al., 2023a) and Ours.

Table 2: Evaluation results for human-centric image generation on hands and faces with controlnet (Zhang et al., 2023a). ($\circ\star$ denotes results evaluated on Laion-5B and UBC-fasion, repectively.)

|        | Faces |  | Hands |  |
|--------|-------|------|-------|------|
| Method | $FID_l$ | $KID_l$ | $FID_l$ | $KID_l$ |
| SD$^\circ$   | 109.57 | 0.0607 | **105.04** | **0.0549** |
| Ours$^\circ$ | **107.98** | **0.0595** | 105.89 | 0.0569 |
| SD$^\star$   | 162.17 | 0.124 | 138.04 | 0.0901 |
| Ours$^\star$ | **154.92** | **0.119** | **130.69** | **0.0890** |

Table 3: Evaluation results on SD 3 with Laion-5B as reference. The results are consistent to performance on SD 1.5 and SD 2, demonstrating the effectiveness of our method across architectures.

| Method | $FID_g$ | $FID_l$ | $KID_g$ | $KID_l$ | CMMD |
|--------|---------|---------|---------|---------|------|
| SD 3 | 15.46 | 25.94 | **0.00384** | 0.00595 | 0.221 |
| Ours | **15.15** | **24.14** | 0.00391 | **0.00503** | **0.157** |

## 6   LIMITATIONS

Our proposed noise re-sampling technique effectively enhances sampling rates and achieves high-fidelity image generation across various scales. However, it encounters limitations in complex interaction scenarios, such as hands engaging with objects. This limitation is rooted in the inherent constraints of latent diffusion models (LDMs), which lack the capability to fully understand complex scenes. Consequently, even with enhanced sampling rates, our method while effective at improving perceptual quality of local patches, can not address the semantic complexities. This highlights the need for approaches that can grasp the dynamics of complex object interactions.

## 7   CONCLUSION

This study has explored the enhancement of latent diffusion models (LDMs) through a novel noise re-sampling strategy, significantly improving the generation quality across different scales. By focusing on the frequency domain and adjusting sampling rates in latent space, our method effectively circumvents the constraints imposed by VAE compression, thereby preserving and enhancing high-frequency details essential for realistic image synthesis. Extensive evaluations demonstrate that our approach outperforms existing methods, including stable diffusion models and super-resolution techniques, particularly in generating detailed and high-fidelity images of complex objects.

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

## A    APPENDIX

### A.1    IMPLEMENTATION DETAIL OF NOISE RE-SAMPLING

We provide pseudo-code for proposed Noise Re-sampling method. This algorithm enhances the detail and texture in specific local regions of an image by adjusting the noise sampling rate, bypassing the constraints of VAE compression and preserving high-frequency information essential for realistic image synthesis.

Table 4: Evaluation Results for human-centric image generation on hands and faces with controlnet Zhang et al. (2023a). Our method demonstrates competitive performance, particularly in enhancing global details. (○ denotes results evaluated on Laion-5B samples and ⋆ denotes experiments on UBC-fasion validation set.)

| Method | Faces | | Hands | |
|---|---|---|---|---|
| | $\text{FID}_g \downarrow$ | $\text{KID}_g \downarrow$ | $\text{FID}_g \downarrow$ | $\text{KID}_g \downarrow$ |
| $\text{SD}^{\circ}$ + Controlnet | 109.35 | 0.057 | **103.16** | **0.0531** |
| $\text{Ours}^{\circ}$ + Controlnet | **109.51** | **0.0585** | 103.033 | 0.0537 |
| $\text{SD}^{\star}$ + Controlnet | 139.92 | 0.102 | 145.44 | 0.0976 |
| $\text{Ours}^{\star}$ + Controlnet | **138.78** | **0.101** | **141.97** | **0.0963** |

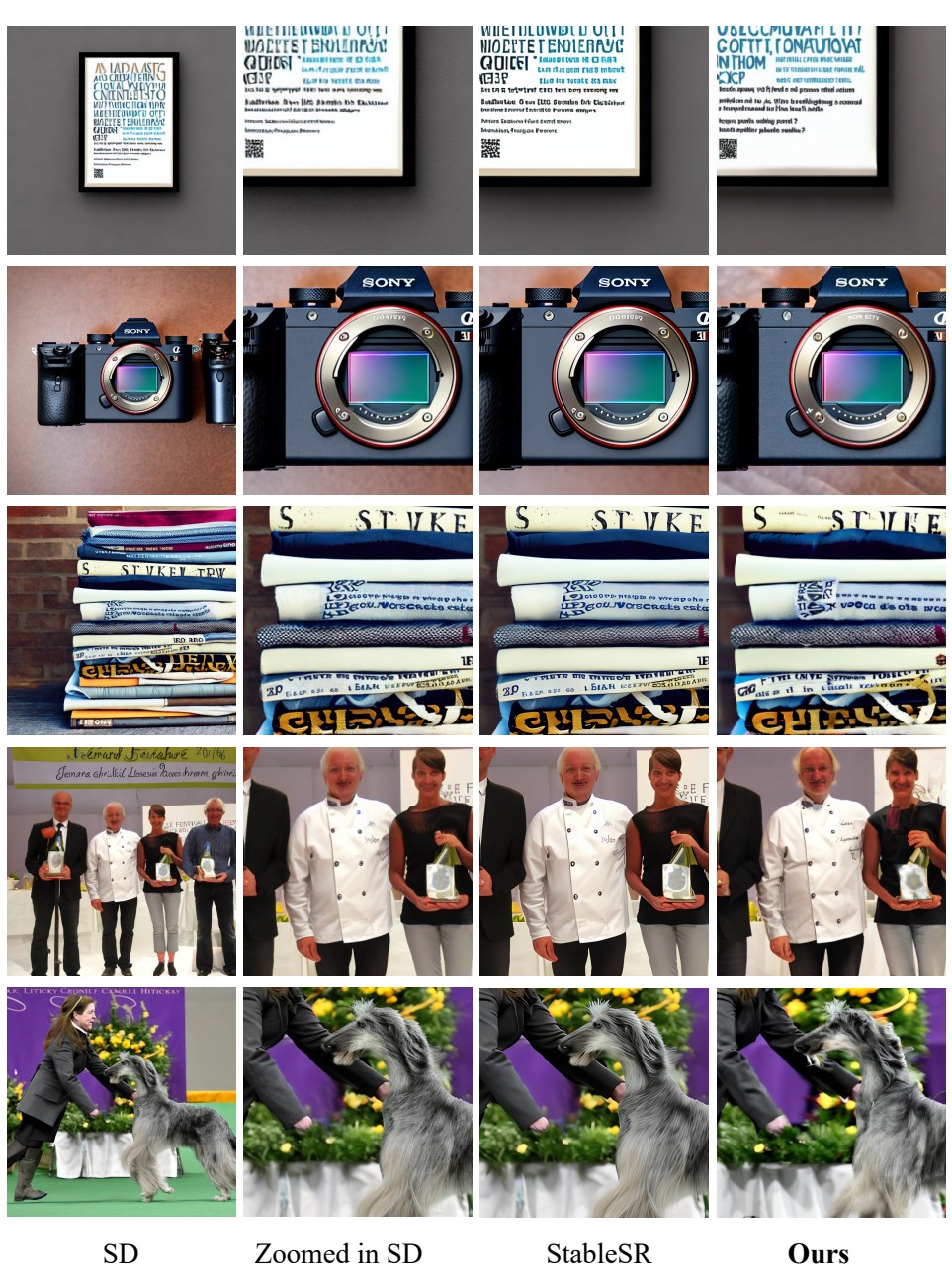

SD        Zoomed in SD        StableSR        **Ours**

Figure 7: Additional visual quality comparisons on SD 2.0 Diffusion (2022).

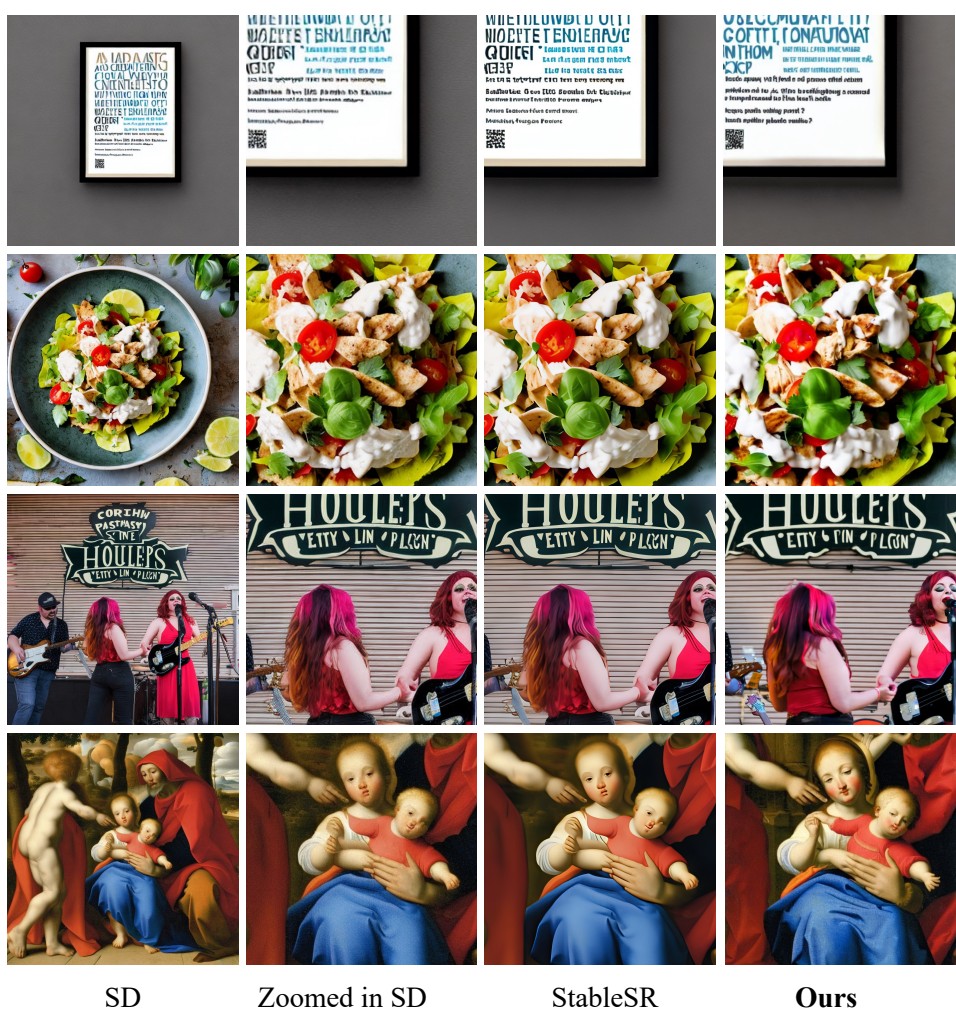

|  SD | Zoomed in SD | StableSR | **Ours** |

Figure 8: Additional visual quality comparisons on SD 2.0 Diffusion (2022).

---

**Algorithm 1** Noise Re-sampling Algorithm

1: **Input:** Initial continuous noise sample $Z_t$, target region $\mathbf{r}$ with size $1/s$, coordinates $(l_x, l_y)$
2: **Input:** $\Delta_s = s/\Delta$ the target sample rate
3: **Output:** High-fidelity clean local latent code $Z_{0,\mathbf{r}}^{\Delta_s}$ with sampling rate $1/\Delta_s$
4: **function** NOISERESAMPLE($Z_t, \mathbf{r}, \Delta_s$)
5: $\quad Z_{t,\mathbf{r}}^{\Delta_s} \leftarrow$ Initialize empty patch
6: $\quad Z_t^{\Delta} \leftarrow$ Initialize empty patch
7: $\quad$ **for** $n \leftarrow 0$ **to** $H - 1$ **do**
8: $\quad\quad$ **for** $m \leftarrow 0$ **to** $W - 1$ **do**
9: $\quad\quad\quad Z_t^{\Delta}[n, m] \leftarrow Z_t[n\Delta, m\Delta]$
10: $\quad\quad\quad Z_{t,\mathbf{r}}^{\Delta_s}[n, m] \leftarrow Z_t[n\Delta_s + l_x, m\Delta_s + l_y]$
11: $\quad\quad$ **end for**
12: $\quad$ **end for**
13: $\quad Z_0^{\Delta} \leftarrow$ Denoise($Z_t^{\Delta}$)
14: $\quad Z_{0,\mathbf{r}}^{\Delta} \leftarrow$ Crop&Upsample($Z_0^{\Delta}, \mathbf{r}$)
15: $\quad \tilde{Z}_{t,\mathbf{r}}^{\Delta_s} = \sqrt{\alpha_t}\mathcal{U}(Z_{0,\mathbf{r}}^{\Delta}) + \sqrt{1 - \alpha_t}Z_{t,\mathbf{r}}^{\Delta_s}$
16: $\quad Z_{0,\mathbf{r}}^{\Delta_s} \leftarrow$ Denoise($\tilde{Z}_{t,\mathbf{r}}^{\Delta_s}$)
17: $\quad$ **return** $Z_{0,\mathbf{r}}^{\Delta_s}$
18: **end function**

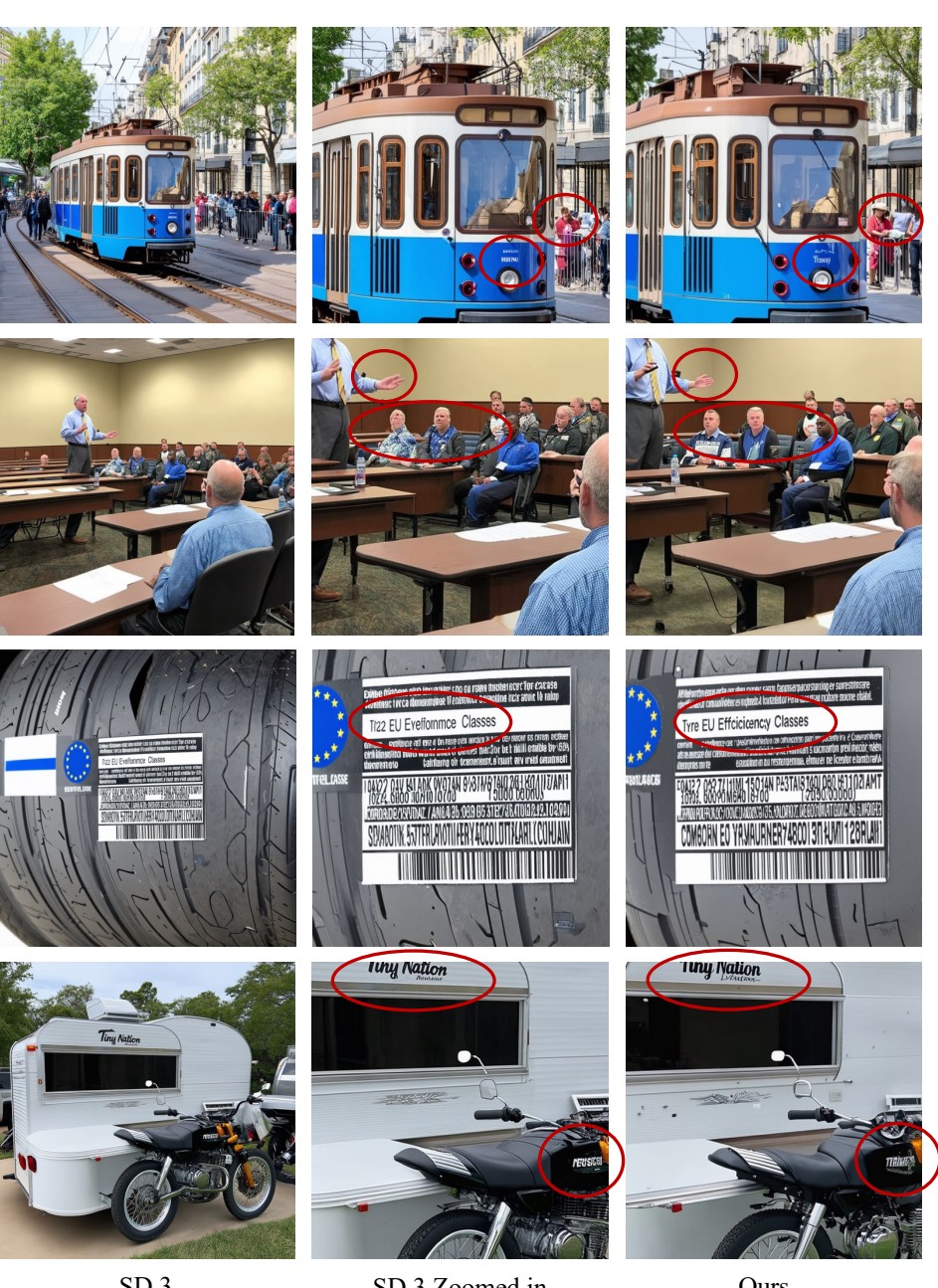

|            |               |      |
|------------|---------------|------|
| SD 3       | SD 3 Zoomed in | Ours |

Figure 9: Visual quality comparisons on SD 3.0 Esser et al. (2024b). Despite its strengths, SD3 often produces artifacts when generating small text, human body or human face that occupies only a small portion of the image. By applying our proposed noise re-sampling approach, these errors can be significantly corrected, resulting in clearer and more accurate generation.

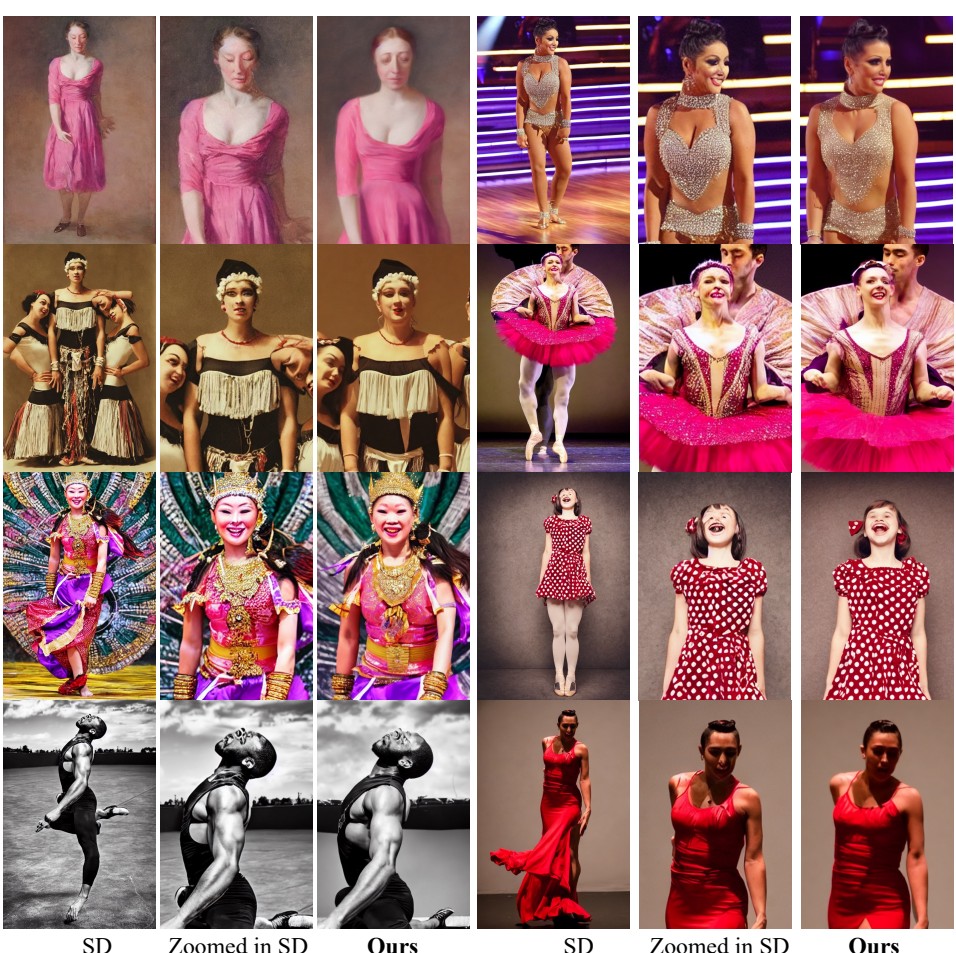

SD     Zoomed in SD     **Ours**        SD     Zoomed in SD     **Ours**

Figure 10: Additional visual quality comparisons on SD 1.5 + Controlnet Diffusion (2022); Zhang et al. (2023a).

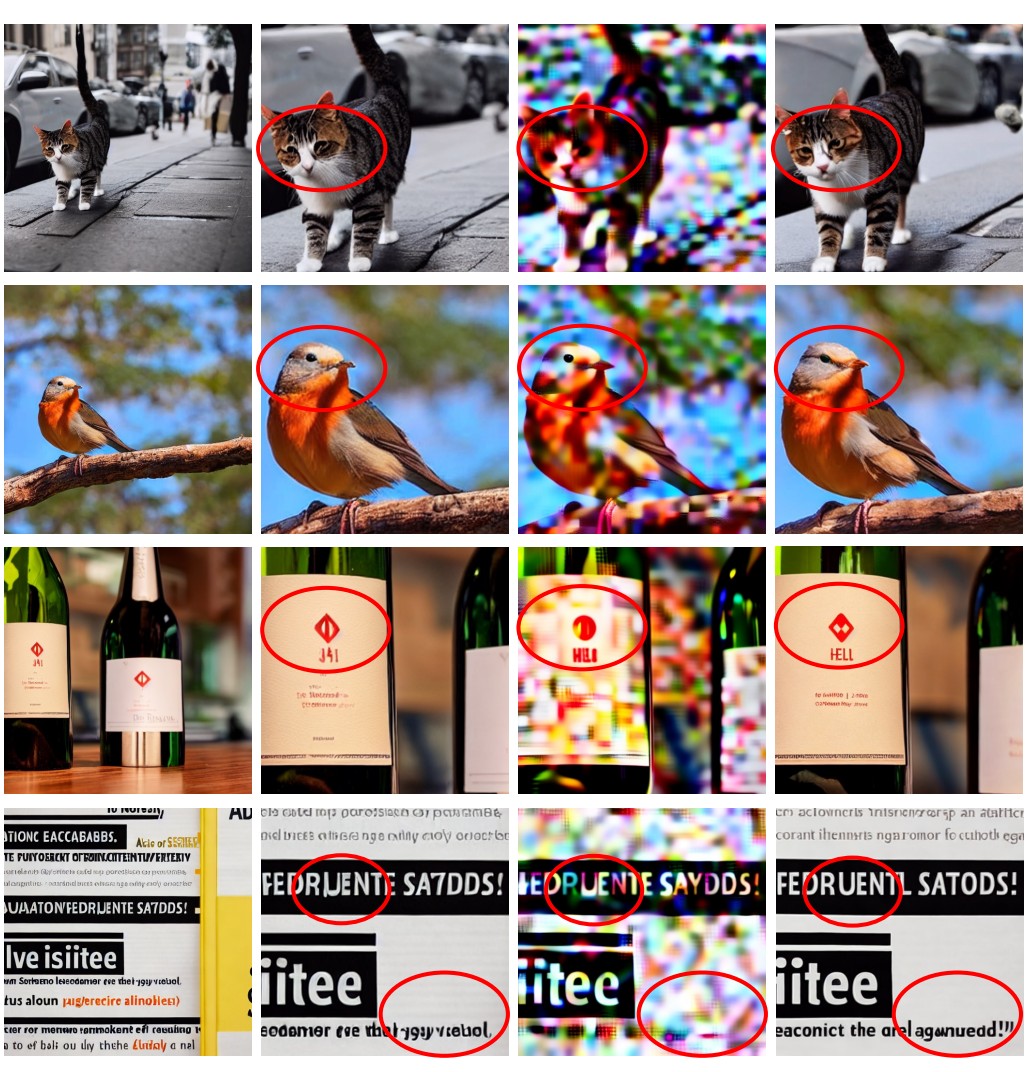

(a) SD 1.5     (b) SD 1.5 zoomed in     (c) Bilinear sampled noise     (d) Ours re-sampled noise

Figure 11: Visualizations comparing bilinearly interpolated noise and our proposed re-sampled noise are shown. As demonstrated in (c), bilinear interpolation distorts the noise distribution, leading to a failure in the denoising process and producing distorted outputs. In contrast, our re-sampled noise preserves the distribution, enabling more accurate denoising and better results.

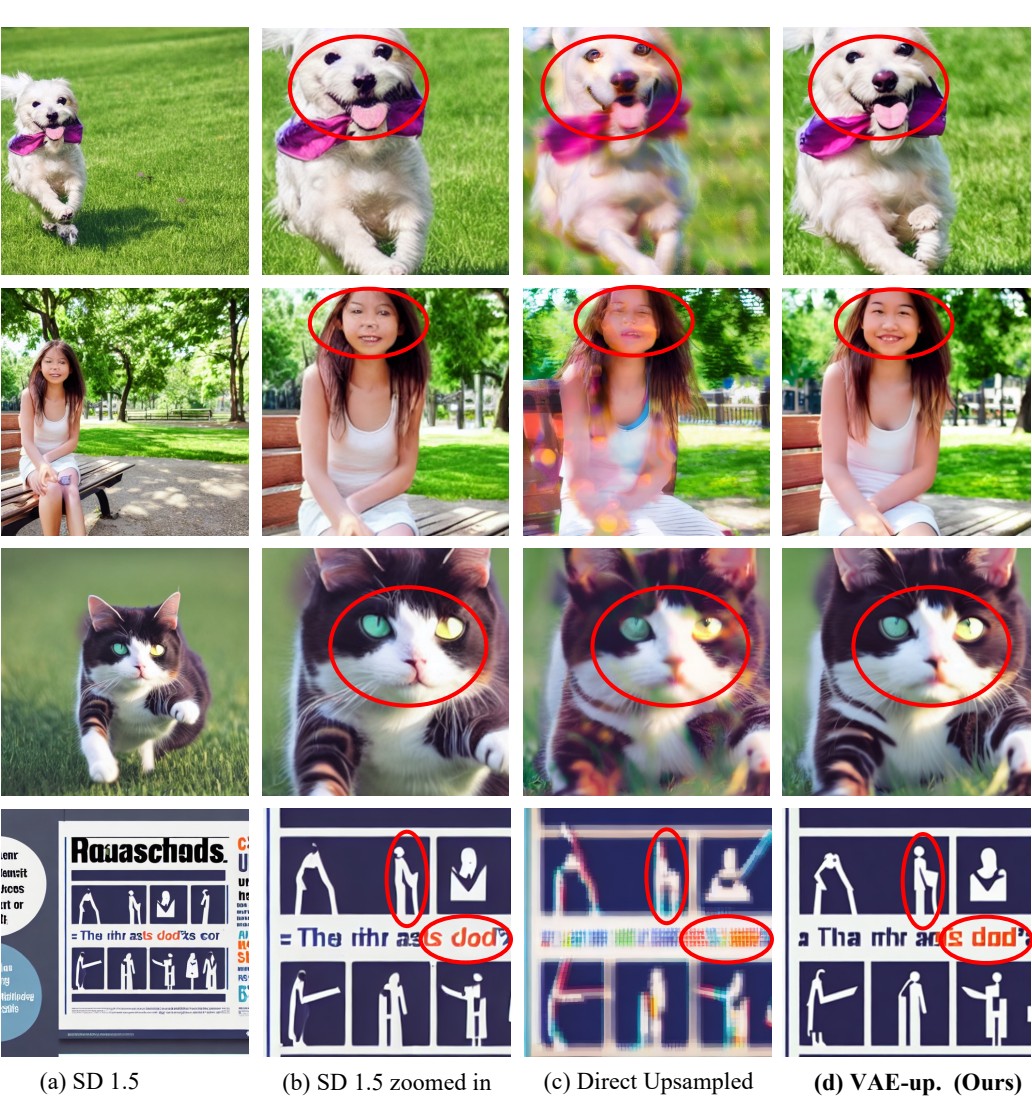

(a) SD 1.5     (b) SD 1.5 zoomed in     (c) Direct Upsampled     **(d) VAE-up.  (Ours)**

Figure 12: Visualizations comparing direct upsampling guidance and our proposed VAE-based upsampling guidance are provided. As shown in (c), direct latent upsampling, when used as guidance, leads to blurry results with color shifts. In contrast, our VAE-based upsampling effectively upsamples the target local region, providing accurate guidance for the re-sampling process and producing results with fine-grained, accurate details.

Table 5: Experiments on SD 2 comparing text-to-image generation task performance with direct latent upsampling and VAE-based upsampling

| Method | $\text{FID}_g \downarrow$ | $\text{FID}_l \downarrow$ | $\text{KID}_g \downarrow$ | $\text{KID}_l \downarrow$ |
|---|---|---|---|---|
| SD 2 | 9.66 | 14.19 | 0.0020 | 0.0033 |
| Ours + Direct Up. | 12.94 | 23.07 | 0.0031 | 0.0086 |
| Ours + VAE-based Up. | **9.56** | **13.94** | **0.0019** | **0.0032** |

## A.2 ADDITIONAL EXPERIMENTS ON HUMAN-CENTRIC IMAGE SYNTHESIS

Additional results were quantitatively assessed using Frechet Inception Distance (FID) and Kernel Inception Distance (KID) metrics, focusing on both global image quality. The improvements in image quality were significant when compared to baseline models, demonstrating our method's ability to preserve and enhance high-frequency details without introducing artifacts or distortions. Qualitatively, the generated images showed remarkable clarity in facial features and hand details, validating the effectiveness of our approach in practical, high-resolution human image synthesis scenarios.

## A.3 ADDITIONAL VISUALIZATION ON TEXT-TO-IMAGE GENERATION TASKS

We have conducted further qualitative comparisons to demonstrate the effectiveness of proposed approach. We selected a diverse set of images and applied our method alongside existing state-of-the-art techniques. Through this comparative analysis, we assessed the visual quality, coherence, and fidelity of the generated samples. By examining various aspects such as texture detail, object clarity, and overall realism, we aimed to provide a comprehensive evaluation of our approach's performance.

## A.4 NOISE SAMPLING FROM SIGNAL PROCESSING VIEW

Assume $Z_t(x, y)$ to be an infinite, continuous 2-D signal in the noise space as shown in Figure 4 (b), which, after denoising, represents an real scene. In practice, we can not directly work on continuous signals. Therefore, current LDMs often starts with a bounded, discrete noise sample $Z_t^\Delta[n, m]$ with sampling period $\Delta$. We reformulate the process of initial noise sampling in the noise space as a bounded discretization process of $Z_t(x, y)$:

$$\delta_\Delta(x, y) = \sum_{h=-\infty}^{\infty} \sum_{w=-\infty}^{\infty} \delta(x - h\Delta)\delta(y - w\Delta), \qquad (9)$$

where $\delta(\cdot)$ is the pulse signal with $\delta(0) = 1$ and 0 elsewhere. $\delta_\Delta$ represents a 2-D pulse train with sampling period $\Delta$. The sampling process can be formulated as:

$$Z_t^\Delta(x, y) = Z_t(x, y)\delta_\Delta(x, y) \qquad (10)$$

To convert the sampled result $Z_t^\Delta(x, y)$ into a bounded discrete space with resolution $H \times W$, we have:

$$Z_t^\Delta[n, m] = Z_t(n\Delta, m\Delta), n \in [0, H], m \in [0, W] \qquad (11)$$

## A.5 ABLATION STUDY ON VAE-BASED UPSAMPLING

As shown in Table 5 and Figure 13, the performance gap under both local and global scenarios justifies our proposed VAE-based upsampling.

## A.6 ADDITIONAL EXPERIMENTS

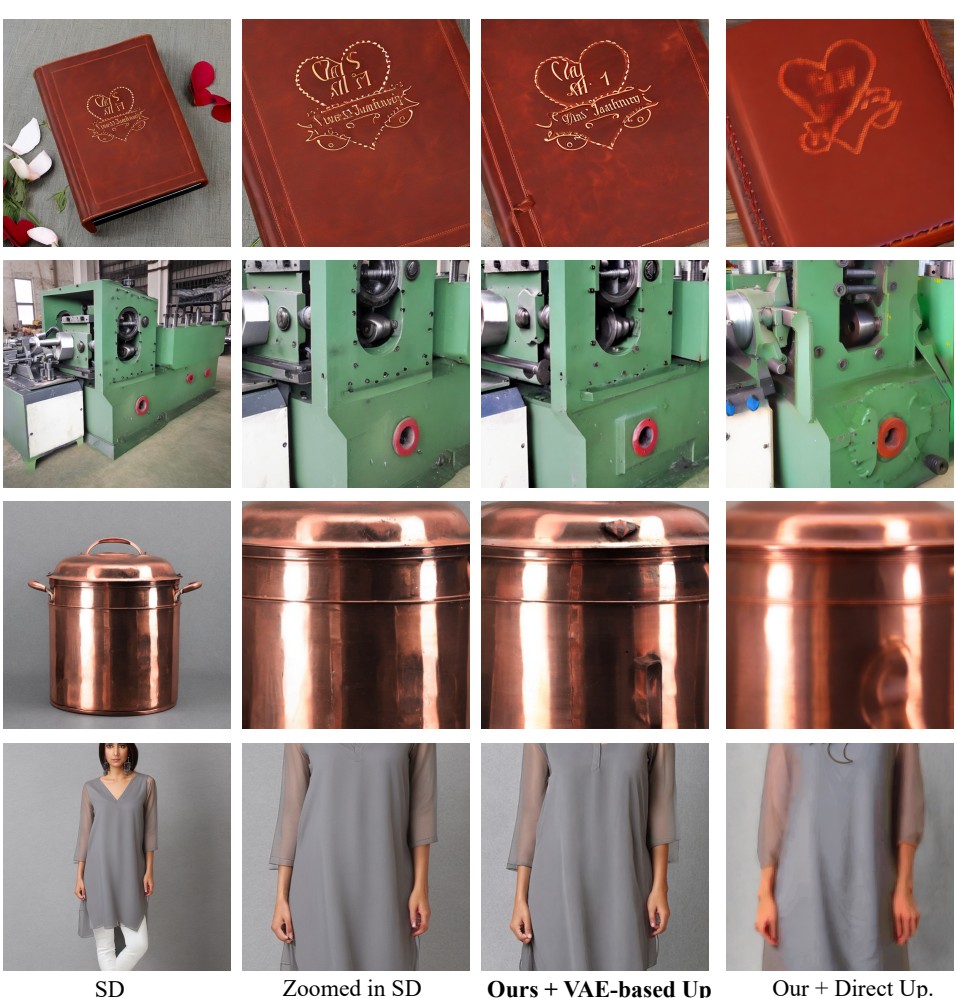

|  |  |  |  |
|---|---|---|---|
| SD | Zoomed in SD | **Ours + VAE-based Up** | Our + Direct Up. |

Figure 13: Abalation study on SD 2 for visual quality comparisons between Direct upsampling and VAE-based Upsampling. Direct upsampling in latent space causes blurred results and inaccurate semantic information.

Table 6: Comparison of computational cost in terms of Gflops and GPU time for different methods.

| Column 1 | Gflops | GPU time (s) |
|---|---|---|
| SD 1.5 | 71,731 | 2.958 |
| Ours (N=25) | 115,287 (60%↑) | 4.627 (56%↑) |
| MultiDiff | 274,200 (282%↑) | 11.292 (281%↑) |
| ScaleCrafter | 279,906 (290%↑) | 16.913 (470%↑) |

Table 7: Evaluation of the hyper-parameter $N$ by measuring the consistency between the original local patch and the re-sampled patch, indicating how well the re-sampled patch aligns with the original local region.

| Re-sampling Step $N$ | 20 | 25 | 30 | 35 | 40 |
|---|---|---|---|---|---|
| Ours ($\text{FID}_l$) | **4.95** | 5.63 | 6.53 | 7.69 | 8.86 |
| Ours ($\text{KID}_l$) | **0.0035** | 0.0026 | 0.0021 | 0.0017 | 0.0014 |

Table 8: Evaluation of the hyper-parameter $N$ on the generation quality of the re-sampled patch, assessing how well the re-sampled local region aligns with the ground truth distribution of the LAION dataset.

| Re-sampling Step $N$ | 20 | 25 | 30 | 35 | 40 |
|---|---|---|---|---|---|
| Ours ($\text{FID}_l$) | 16.41 | 14.03 | 14.52 | 13.77 | **12.60** |
| Ours ($\text{KID}_l$) | 0.0044 | 0.0035 | 0.0028 | 0.0027 | **0.0024** |

Table 9: Experiments evaluating the impact of the local ratio $s$ on the quality of re-sampled patches. Ratios between 0.33 and 0.67 strike the optimal balance, while ratios outside this range lead to decreased performance.

| Local Ratio $s$ | vanilla SD1.5 | 0.67 | 0.50 | 0.33 | 0.25 | 0.20 |
|---|---|---|---|---|---|---|
| Ours ($\text{FID}_l$) | 15.60 | **12.82** | 14.03 | 21.21 | 26.53 | 32.38 |
| Ours ($\text{KID}_l$) | 0.0044 | **0.0026** | 0.0032 | 0.0047 | 0.0062 | 0.0083 |

Table 10: Comparison of different methods, including Consistency Decoder (CD) and Guidance Interval (GI), to evaluate their impact on generation quality.

| Method | SD1.5 | Ours | CD | CD + Ours | GI | GI + Ours |
|---|---|---|---|---|---|---|
| $\text{FID}_l$ | 15.60 | **14.03** | 16.33 | 15.40 | 32.17 | 26.49 |
| $\text{KID}_l$ | 0.0044 | 0.0035 | 0.0034 | **0.0031** | 0.0183 | 0.0114 |

**Ablation study on re-sampling step** $N$    In Table 7, we evaluate the consistency between the original local patch and the re-sampled patch by measuring $\text{FID}_l$ and $\text{KID}_l$, which reflect how well the re-sampled patch aligns with the original local region. In Table 8, we measure how well the re-sampled local region matches the ground truth distribution of the LAION dataset, again using $FID_l$ and $KID_l$ to quantify this alignment. From 7, we observe that smaller $N$ values provide better consistency with the original local patch, as indicated by lower $FID_l$ and $KID_l$ values. In Table 8, larger $N$ values result in better alignment with the ground truth distribution, as seen by lower $FID_l$ and $KID_l$ values. These results indicate a trade-off between consistency with the original local patch and alignment with the global ground truth distribution, offering insights into how $N$ can be tuned to balance these factors effectively. In our experiments, $N$ is set to half of the total denoising steps $T$ to trade off between consistency and quality.

**Ablation study on re-sampling ratio** $s$    In Table 9, we examine how the local ratio $s$ affects the quality of re-sampled patches. From the results, it is evident that a local ratio between 0.33 and 0.67

achieves the best balance, leading to significant improvements in the generation quality of the local regions. Ratios outside this range tend to result in diminished performance worse than vanilla SD, as indicated by higher $FID_l$ and $KID_l$ values.

**Comparison to Consistency Decoder and Guidance Interval** Quantitative results consistently demonstrate the superiority of our approach in handling small or complex regions. Quantitative metrics show that our method achieves better scores in local enhancement.

Additionally, we emphasize that our method is not mutually exclusive with the Consistency Decoder (Song et al., 2023) and Guidance Interval (Kynkäänniemi et al., 2024). In fact, the two approaches can be seamlessly integrated to combine their strengths. By applying our method to enhance the latent representation before decoding and then using the consistency decoder for final image reconstruction, it is possible to achieve even greater visual fidelity and detail enhancement as shown in Table 10.

