# OpenReview forum: "Noise Re-sampling for High Fidelity Image Generation"
_ICLR.cc/2025/Conference — Submitted to ICLR 2025_

### Official Review · Reviewer_92CH · 2024-10-22

**Soundness:** 2
**Presentation:** 1
**Contribution:** 2
**Rating:** 6
**Confidence:** 4

**Summary:**

This paper proposes a noise re-sampling strategy and VAE-based upsampling to address the lossy compression problem of VAE in latent diffusion models, which are almost essential in the text-to-image field. The authors observe distortions in small objects generated by latent diffusion models like Stable Diffusion 1, 2, and 3, and propose increasing the sampling rates of local regions. The proposed noise re-sampling method crops the local region of the latent image generated by the diffusion model, upscales it using the proposed VAE-based upsampling method to match the size learned by the diffusion model, and then refines the upsampled crop before attaching it to the original image. The paper compares this method with approaches like Stable SR and Multi-Diffusion, demonstrating superior qualitative and quantitative performance.

**Strengths:**

- This paper highlights the issue that VAE compression errors are more prominent in lower-resolution images, a problem that the generative AI community has not paid close enough attention to.

- The proposed method is training-free, making it easy to apply with minimal effort to any newly developed latent diffusion model.

**Weaknesses:**

- The most fundamental way to address the VAE's lossy compression issue is by using a better VAE. For instance, OpenAI's consistency decoder [1] replaces the decoder of Stable Diffusion 1's VAE with a new diffusion model and releases model weights that can rapidly decode latents into images via consistency distillation [2]. Samples from the consistency decoder demonstrate strong reconstruction performance, particularly on low-resolution (256x256) images where reconstruction errors, like those mentioned in this paper, often occur. This paper should also include a comparison with the consistency decoder.

- Another approach is to ensure the diffusion model generates better latents. Recent research on diffusion guidance [3] suggests that strong classifier-free guidance at middle noise levels enhances image detail. The proposed noise re-sampling method also requires experiments with various guidance scales.

- Some implementation details are missing. The local patch chosen for upsampling should ideally be a small object prone to distortion, but how is this patch selected? What is the exact upsampling ratio used in experiments? When denoising the local patch with the diffusion model, what text prompt is used?

- The proposed VAE-based upsampling seems counterintuitive because it relies on the same VAE that the paper argues to suffer lossy compression issues. This suggests that the upsampled patch might contain significant artifacts.

- The paper lacks of qualitative results of SD3 and SDXL.

[1] Open AI, https://github.com/openai/consistencydecoder

[2] Song et al., Consistency Models, ICML 2023.

[3] Kynkäänniemi et al., Applying Guidance in a Limited Interval Improves Sample and Distribution Quality in Diffusion Models, 2024.

**Questions:**

Stable Diffusion 3 is known for its ability to generate text well. Can noise re-sampling correct errors when text generation goes wrong?

---

> ### Author Response · Authors · 2024-11-25
> **Response to reviewer 92CH**
>
> Dear reviewer 92CH,
>
> Thank you for taking the time to review our work. We deeply appreciate your feedback and have responded to your concerns as outlined below:
>
> ```properties
> 1. The most fundamental way to address the VAE's lossy compression issue is by using a better VAE. For instance, OpenAI's consistency decoder [1]...This paper should also include a comparison with the consistency decoder.
> ```
> Thank you for the comment. We acknowledge the value of the consistency decoder as an enhancement to the standard VAE decoder in improving sharpness and reconstructing missing details. However, it is important to **highlight the distinctions** between the two approaches and their respective strengths, as well as how they can **complement each other**.
>
> The consistency decoder operates similarly to **super-resolution-based methods**, primarily enhancing the perceptual quality of decoded images by **sharpening details and "guessing" missing information.** While this improves the visual fidelity of the output, it does **not** fundamentally address the **artifacts caused by limited sampling rates in the latent space**. These artifacts are deeply rooted in the resolution constraints imposed during the generation process, particularly for complex or small-scale details. By focusing solely on the image space, the consistency decoder does not directly resolve the loss of high-frequency information that occurs at the latent level.
>
> Our method takes a fundamentally different approach by addressing the issue of limited sampling rates directly in the latent space. By introducing the concept of sampling rate into the denoising process, we enhance the resolution of local regions in the latent representation itself, allowing for a more **robust recovery of fine-grained details**. This approach ensures that the improvements occur at the core of the generative process, rather than relying on post-hoc enhancements in the decoded image.
>
> To ensure a comprehensive evaluation, we have included comparisons with the consistency decoder in our experiments (CD indicates Consistency decoder. GI for Guidance Interval[1]).
> | Method  | SD1.5 | SD1.5 + Ours | SD1.5 + CD | SD1.5 + CD + Ours | SD1.5 + GI | SD1.5 + GI + Ours |
> | ------- | ----| ----- | ---------- | ----------------- | ---------- | ----------------- |
> | $FID_l$ | 15.60 | **14.03**  |  16.33  |  15.40           | 32.17   |   26.49       |
> | $KID_l$  |   0.0044 | 0.0035  |  0.0034  |    **0.0031**      |  0.0183    |        0.0114  |
>
> Quantitative results consistently demonstrate the superiority of our approach in handling small or complex regions. Quantitative metrics show that our method achieves better scores in local enhancement.
>
> Additionally, we emphasize that our method is not mutually exclusive with the consistency decoder. In fact, the two approaches can be **seamlessly integrated** to combine their strengths. By applying our method to enhance the latent representation before decoding and then using the consistency decoder for final image reconstruction, it is possible to achieve even greater visual fidelity and detail enhancement as shown in the table.
>
> ```properties
> 2. Another approach is to ensure the diffusion model generates better latents. Recent research on diffusion guidance [3] suggests that strong classifier-free guidance at middle noise levels enhances image detail. The proposed noise re-sampling method also requires experiments with various guidance scales.
> ```
> We appreciate the reviewer’s suggestion regarding the exploration of diffusion guidance and its relevance to noise re-sampling. First, the study in [3] investigates how applying guidance in specific intervals, rather than throughout the entire denoising process, affects the generation process and enhances image detail. While this is an interesting approach, it primarily focuses on optimizing the distribution of guidance over time rather than addressing structural inconsistencies or compression-induced losses, which are the focus of our noise re-sampling method.
>
> Secondly, we have added a comparison with Guidance Interval (GI) in the revised version and attached the table above. However, its performance is limited. Notably, the original paper [3] does not provide quantitative comparisons, making it difficult to comprehensively evaluate its efficacy. Additionally, **our method is complementary to Guidance Interval and can be applied concurrently**. This combined approach could further enhance generation quality by leveraging both techniques simultaneously, addressing structural and detail fidelity issues from different perspectives.
>
> [3] Kynkäänniemi et al., Applying Guidance in a Limited Interval Improves Sample and Distribution Quality in Diffusion Models, 2024.

---

> ### Author Response · Authors · 2024-11-25
> **Response to reviewer 92CH (2)**
>
> ```properties
> 3. The local patch chosen for upsampling should ideally be a small object prone to distortion, but how is this patch selected?
> ```
> In our experiments, local patches were selected **randomly to demonstrate the general applicability** of our approach. In real-world applications, the selection process can be **customized to focus on regions more prone to distortion or containing critical details**, such as small objects or intricate textures. While automated methods for detecting such artifacts would be highly desirable, to the best of our knowledge, no such method currently exists specifically for identifying artifacts in LDM-generated results. Developing an automated artifact detection mechanism would be a valuable direction for future research, further enhancing the practicality and effectiveness of artifact correction.
>
> ```properties
> 4. What is the exact upsampling ratio used in experiments?
> ```
> For convenience, the upsampling ratio used in our experiments is fixed at 0.5, with the local region randomly located and covering an area that is 0.25 times the size of the original image. Additionally, we have included an ablation study in the revised version to demonstrate how varying the upsampling ratio impacts performance, providing insights into the trade-offs and effectiveness of different configurations. We attach the table here for easy reference.
>
> | Local Ratio $s$ | vanilla SD1.5 | 0.67  | 0.50  | 0.33  | 0.25    | 0.20    |
> | -------------------- | --- | ----- | ----- | --- | ----- | ----- |
> | Ours  ($FID_l$)      | 15.60 |  **12.82** | 14.03 |  21.21  | 26.53 | 32.38 |
> | Ours ($KID_l$)       | 0.0044 |  **0.0026**  |  0.0032  |   0.0047  | 0.0062 |   0.0083    |
>
> ```properties
> 5. When denoising the local patch with the diffusion model, what text prompt is used?
> ```
> The text prompt used for denoising the local patch is the same as that used for the global image. During our experiments, we observed that since the re-sampling process begins at later stages of denoising, the influence of the text prompt on the semantics of the re-sampled local region is significantly diminished, which is also supported by recent work [1]. This ensures consistency between the local patch and the global context without introducing semantic deviations.
>
> [1] Yi et al, Towards Understanding the Working Mechanism of Text-to-Image Diffusion Model. NeurIPS 2024.
>
> ``` properties
> 6. The proposed VAE-based upsampling seems counterintuitive because it relies on the same VAE that the paper argues to suffer lossy compression issues. This suggests that the upsampled patch might contain significant artifacts.
> ```
> We appreciate the reviewer’s concern regarding the use of VAE-based upsampling despite the lossy nature of VAEs. It is important to clarify that the primary purpose of VAE-based upsampling is to **avoid the issues caused by directly applying upsampling in the latent space**, which **distorts the latent statistics and often leads to failure in the denoising process**. By leveraging the VAE, the upsampling operation is moved to the image space, where it may indeed introduce some inaccuracies. However, these inaccuracies are mitigated when the upsampled image is encoded back into the latent space through VAE compression, which removes the majority of such artifacts.
>
> Moreover, the subsequent denoising process, operating with corrected latent statistics, has been observed to be robust against the residual inaccuracies introduced by VAE-based upsampling. **Empirically, this approach outperforms direct latent-space upsampling, as evidenced by improved results in both visual quality and quantitative metrics**. Nevertheless, we acknowledge that the statistical deviations caused by latent-space upsampling remain an interesting area for further investigation and refinement in future work.
>
>
> ```properties
> 7. The paper lacks of qualitative results of SD3 and SDXL.
> ```
> Due to time constraints, we have included qualitative results for SD3 in the appendix (Figure 9). If additional time is available, we would include results for SDXL in a subsequent revision. By applying our method, we significantly improve SD 3's abillity at generating objects (e.g., hands, faces and texts) that are complex yet small in the image, demonstrating the generalization ability of proposed approach.
>
>
> ```properties
> 8. Stable Diffusion 3 is known for its ability to generate text well. Can noise re-sampling correct errors when text generation goes wrong?
> ```
> We have added a qualitative comparison for SD3 in the revised version (See Appendix Figure 9). Despite its strengths, SD3 often produces artifacts when generating small text that occupies only a small portion of the image. By applying our proposed noise re-sampling approach, these errors can be significantly corrected, resulting in clearer and more accurate text generation.

---

> > ### Comment · Reviewer_92CH · 2024-11-27
> > **VAE-based upsampling still seems counterintuitive**
> >
> > I appreciate the results showing that the proposed method is effective when used with the Consistency decoder and guidance interval. However, I still have concerns about the VAE-based upsampling, which appears counterintuitive. The upsampling ratio ablation study in the rebuttal shows that the method is sensitive to the sampling rate. As the upsampling becomes more aggressive, performance rapidly deteriorates, and when $s$ is smaller than 0.5, the results are even worse than the vanilla text-to-image generation. I believe this outcome is due to error accumulation caused by the VAE-based upsampling. Furthermore, the explanation in the rebuttal that VAE encoding removes artifacts also seems counterintuitive, as the VAE encoder might amplify these artifacts if they are present in the image.

---

> ### Author Response · Authors · 2024-11-27
> **Response to reviewer 92CH on concerns about VAE-Based Upsampling:**
>
> Thank you for your detailed response and for raising these important points. We appreciate the opportunity to address your concerns further.
>
> 1. **Clarification on "vanilla SD 1.5"**: We apologize for the misleading entry "vanilla SD 1.5." It was intended as a reference for the generation quality of SD 1.5 at a zoom-in ratio of 0.5, rather than an indicator of its performance at all ratios. To provide additional clarity, we have included the generative quality metrics of vanilla SD 1.5 across different ratios for your reference. As demonstrated, **our proposed approach consistently provides solid improvements over SD 1.5 at each ratio**. Additionally, regarding the degradation observed with respect to the upsampling ratio, we would like to emphasize that this is not solely "due to error accumulation caused by the VAE-based upsampling." Rather, it is an **inherent limitation of current upsampling methods such as bilinear or bicubic interpolation**, which are widely used for image resizing. These methods **inherently introduce artifacts and distort the image as the ratio increases, especially when applied to latent space representations**. As the upsampling becomes more aggressive, the distortion in the latent distribution increases, which can cause a decline in image quality. Despite this, our proposed method managed to mitigate this artifacts and improve the generation quality at each ratio.
>
> | Local Ratio $s$   | 0.67       | 0.50       | 0.33       | 0.25       | 0.20       |
> | ----------------- | ---------- | ---------- | ---------- | ---------- | ---------- |
> | SD 1.5 ($FID_l$)  | 13.08      | 15.60      | 26.36      | 36.72      | 46.96      |
> | Ours  ($FID_l$)   | **12.82**  | **14.03**  | **21.21**  | **26.53**  | **32.38**  |
> | SD 1.5  ($KID_l$) | 0.0029     | 0.0044     | 0.0068     | 0.0108     | 0.0153     |
> | Ours ($KID_l$)    | **0.0026** | **0.0032** | **0.0047** | **0.0062** | **0.0083** |
>
> 2. **Purpose of VAE-based upsampling**: We would like to clarify that the primary purpose of VAE-based upsampling is to provide an **imperfect, upsampled latent code with *consistent semantic information* for the re-sampling process**. The VAE-based upsampling does not aim to perfectly upsample the latent code, but rather serves to provide a sufficient approximation that guides the re-sampling process. The re-sampling process itself is the key method we designed to remove artifacts and restore fine-grained details. To support this, we have provided visualizations in Figures 11 and 12 of the appendix, which illustrate the consequences of directly applying latent upsampling. These figures show that **directly upsampling the latent codes distorts their statistical properties (color shifts, semantic losses and etc.), leading to inaccurate guidance for the re-sampling process**. In contrast, the VAE-based upsampling helps preserve semantic consistency while avoiding the issues caused by direct latent upsampling.
>
> 3. **VAE-base upsampling is a practical solution for conducting upsampling in the latent space** without significantly distorting the latent distribution or causing the denoising process to fail. This approach allows us to **avoid introducing additional modules or excessive computational costs**. To the best of our knowledge, we are the first to systematically study the effects of latent upsampling, and currently, there is **no better alternative than using the VAE**. While the encoding process in the VAE may compress artifacts into the latent space, as demonstrated in the appendix, it **provides sufficiently accurate guidance to our proposed re-sampling process**, producing final accurate local regions with fine-grained details, which remains the central contribution of this work.
>
> We hope this explanation addresses your concerns and provides a clearer perspective on the role of VAE-based upsampling and the contributions of our approach. If you have any additional questions or suggestions, we would be happy to address them. Thank you again for your thoughtful feedback.

---

> > ### Comment · Reviewer_92CH · 2024-11-27
> >
> > Thank you for clarifying the table in the rebuttal. The authors have addressed all my concerns, and I will raise my score from 3 to 6.
> >
> > However, I strongly recommend updating the qualitative results for SDXL as well.

---

> > > ### Author Response · Authors · 2024-11-29
> > > **Response to Reviewer 92CH’s Feedback and SDXL Results Update**
> > >
> > > Thank you for raising the score and for your constructive feedback. We are glad to hear that the clarifications in the rebuttal have addressed your concerns.
> > >
> > > Regarding the SDXL results, we will certainly include them if a revision is possible. We appreciate your suggestion and will prioritize adding these results in the revised version.
> > >
> > > Thank you again for your time and thoughtful consideration.

---

### Official Review · Reviewer_VJ6f · 2024-10-26

**Soundness:** 3
**Presentation:** 3
**Contribution:** 2
**Rating:** 5
**Confidence:** 4

**Summary:**

The paper proposes a Noise Re-sampling strategy to enhance high-fidelity image generation in Latent Diffusion Models (LDMs). This method selectively increases sampling rates in detailed local regions. Additionally, a VAE-based upscaling technique ensures consistency across local and global regions by upscaling latent spaces in the image domain. Experiments shows that the approach outperforms existing super-resolution and high-resolution adaptation methods on SD backbones.

**Strengths:**

1. The paper introduces the concept of Noise Resampling, which is somewhat novel to me in the context of latent diffusion.
2. The paper studies the error in VAE reconstruction and attempts to address it with noise resampling. It's somewhat valuable.
3. The paper is well organized.

**Weaknesses:**

1. The proposed solution is straightforward. To address the low-quality fine-grained details in SD generation, the method adds noises to generated patches then conduct the backward denoising process again. Although some methods such as VAE upscaling has been proposed to improve this method, this idea is straightforward without many insights into the diffusion process.

2. The proposed method requires massive denoising and VAE upsampling over different resolutions of the generation. This would introduce significant computational costs. Unfortunately, I didn't see any discussions on method efficiency in the paper.

3. The performance improvement on global metrics, such as FID and KID, is trivial.

**Questions:**

See above

---

> ### Author Response · Authors · 2024-11-25
> **Response to reviewer VJ6f**
>
> Dear reviewer VJ6f,
>
> We sincerely appreciate your thorough evaluation of our work. Below, we respond to the concerns you’ve outlined:
>
> ```properties
> 1. Although some methods such as VAE upscaling has been proposed to improve this method, this idea is straightforward without many insights into the diffusion process.
> ```
> Thank you for the feedback. We would like to elaborate on how our method redefines the utilization of Latent Diffusion Models (LDMs) by addressing their limitations through the integration of the sampling rate concept and the development of a synergistic resampling process.
>
> At the core of our work is the recognition that LDMs, while powerful, face **inherent constraints in generating small or complex objects** due to their fixed-resolution design and **limited native sampling rates**. These limitations restrict their ability to fully capture high-frequency details, leading to noticeable deficiencies in fine-grained regions. By **introducing the concept of sampling rate into the denoising process**, we provide a new perspective on the generative capacity of LDMs, allowing us to **quantify and address their limitations** more systematically.
>
> Our resampling process builds directly on this insight, redefining how LDMs can be leveraged to overcome their native constraints. Rather than treating denoising as a static or fixed-resolution operation, we introduce a **dynamic approach that adapts the sampling rate for local regions requiring enhanced detail**. This process works synergistically with the LDM’s generative strengths by leveraging its native capabilities at fixed resolution while augmenting these with higher sampling rates in targeted areas. By combining the local resampling with a reference derived from the global image, we ensure that the process remains **consistent** with the overall structure and context of the image.
>
> What sets this approach apart is its ability to extend the capabilities of LDMs **without disrupting their inherent strengths**. The resampling process integrates seamlessly into the iterative denoising framework, using the model’s existing architecture and pre-trained features to enhance high-frequency details. This avoids the pitfalls of alternative methods, such as simple upscaling which can introduce inconsistencies or artifacts, or super-resolution methods which sharpens the details but failed to correct artifacts.
>
> ```properties
> 2. The performance improvement on global metrics, such as FID and KID, is trivial.
> ```
> We appreciate the reviewer’s observation regarding the performance improvement on global metrics such as FID and KID. However, we would like to clarify the focus and intent of our method, which emphasizes **local enhancement**, and explain how this focus could limits its impact on global metrics.
>
> Our method is explicitly designed to address the challenges of generating **fine-grained details in localized regions** of an image, particularly for **small or complex objects**. In our experimental evaluation, the targeted local regions typically occupy **less than 25% of the overall image area**. Consequently, even substantial improvements in these regions would have a limited effect on global metrics like FID and KID, which assess the entire image holistically.
>
> Furthermore, it is important to note that both FID and KID rely on pretrained vision backbones that perform **significant downsampling**, often by a factor of up to 16 times [1][2]. This downsampling aggregates information across large regions of the image, effectively **diminishing the impact of localized enhancements**. As a result, the improvements introduced by our method in small, detailed regions **may not be adequately reflected in these global metrics.**
>
> To address this limitation, we also evaluated our method using localized metrics, which are specifically designed to **assess the quality of small, targeted regions within an image**. These metrics more directly capture the improvements achieved by our approach, and the results clearly demonstrate its effectiveness in enhancing the **quality of local regions**.
>
> In summary, while the impact of our method on global metrics like FID and KID may appear trivial due to the inherent limitations of these metrics in capturing local detail enhancements, the localized metrics provide a more accurate and comprehensive evaluation of our method’s performance. This focus on local enhancement aligns with the core objective of our work and highlights the significance of the improvements achieved in regions that matter most for high-fidelity image generation.
>
> [1] Simonyan, K. "Very deep convolutional networks for large‐scale image recognition." Proc ICLR (2015).
>
>
> [2] Radford, Alec, et al. "Learning transferable visual models from natural language supervision." International conference on machine learning. PMLR, 2021.

---

> ### Author Response · Authors · 2024-11-25
> **Response to reviewer VJ6f (2)**
>
> ```properties
> 3. The proposed method requires massive denoising and VAE upsampling over different resolutions of the generation. This would introduce significant computational costs. Unfortunately, I didn't see any discussions on method efficiency in the paper.
> ```
> We appreciate the reviewer’s concern about inference speed and recognize the importance of computational efficiency. To address this, we provide a detailed comparison and added to the revised appendix. Our method increases Gflops by 60% and GPU time by 56% compared to SD 1.5, but it is **far more efficient than existing methods** like MultiDiff and ScaleCrafter, which exhibit significantly higher computational overhead. These results demonstrate that our approach effectively balances improved image quality with computational efficiency. We attach the table here for easy reference.
> | Column 1                     | Gflops                   | GPU time (s)             |
> | ---------------------------- | ------------------------ | ------------------------ |
> | SD 1.5                       | 71,731                   | 2.958                    |
> | Ours (N=25)                  | 115,287 (60%$\uparrow$)  | 4.627 (56%$\uparrow$)    |
> | MultiDiff | 274,200 (282%$\uparrow$) | 11.292  (281%$\uparrow$) |
> |  ScaleCrafter       |    279,906 (290%$\uparrow$)  |  16.913 (470%$\uparrow$)   |

---

> ### Author Response · Authors · 2024-11-29
> **Follow-up to Reviewer VJ6f**
>
> Dear Reviewer VJ6f,
>
> Thank you again for your detailed feedback. We have carefully addressed your concerns in our previous response and revised the manuscript. In summary:
>
> 1. **Connection to Diffusion Process**: We clarified how our approach is directly linked to the diffusion process. We explored the inherent limitations of current diffusion models in generating small or complex objects due to their fixed-resolution design and limited native sampling rates. By **studying the properties of noise and latent distributions**, we highlighted why traditional upsampling methods (e.g., bilinear, bicubic) are not suitable for the latent space in diffusion models. Our proposed noise re-sampling, based on these insights, adapts LDMs to enhance fine-grained details without introducing artifacts or inconsistencies through **aligned global & local noise distribution consistency and guided re-sampling**, ultimately **unleashing the full generative potential of current LDMs**.
>
> 2. **Performance on Global Metrics (FID and KID)**: While improvements in global metrics are not substantial due to our focus on local regions, we demonstrated significant gains using localized metrics that better capture enhancements in small, targeted areas.
>
> 3. **Computational Efficiency**: We provided a detailed comparison showing that our method remains significantly more efficient than methods like MultiDiff and ScaleCrafter.
>
> We hope these clarifications address your concerns. If all issues are resolved, we would be grateful if you could kindly consider raising your score.
>
> Thank you again for your time and consideration.

---

### Official Review · Reviewer_9GdS · 2024-11-02

**Soundness:** 3
**Presentation:** 3
**Contribution:** 3
**Rating:** 6
**Confidence:** 4

**Summary:**

This work focuses on the noise sampling rate/resolution in LDMs, and proposes a local reference re-sampling strategy to improve the high-resolution image generation performance.
The proposed method uses the normal global noise-based denoised image as the reference, to further guide another branch which is focused on local region.
Meanwhile, the method deploys a training-free mixed upsampling strategy to provide corresponding local latent noise aiming at generating patches with better quality.
Extensive experiments show the proposed method achieves good performance, especially for complex objects and other local details.

**Strengths:**

1. The idea is quite simple but overall I think it is a good paper. Instead of directly increasing the sampling rate, using VAE to perform local space re-sampling in the image space is quite interesting, as it avoids the need for a higher-resolution dataset and re-training. Figure 5 also demonstrates that this method indeed captures more high-frequency information.

2. The proposed re-sampling strategy is straightforward and can be plug-and-play for LDMs.

3. The author's exploration of the impact of VAE compression on the generation quality of LDMs aligns with intuition, and this paper reminds me of a work IDM [1] that uses cascaded Diffusion Models for continuous super-resolution

5. The authors provide a demo code which is worth encouraging. I've tried it and it works fine.

[1] Gao S, Liu X, Zeng B, et al. Implicit diffusion models for continuous super-resolution[C]. CVPR 2023.

**Weaknesses:**

1. Figure 4 is kind of ambiguous.

     (1): If I understand correctly, the blue boxes in subfigures (a) and (b) should be the same size. I also suggest indicating the size of each box for clarity.

     (2): I suggest removing the "Direct latent space upsampling" branch below subfigure (c), or replacing it with the noise re-sampling branch corresponding to Figure 5 (d). Additionally, the title for subfigure (c) appears twice.

2. The statement in lines 367-369, "The final re-sampled clean latent code can be directly pasted back to the corresponding local region of the original output after decoding" is ambiguous. Based on Figure 4 (a), it should be decoded first, and then pasted back to the output image.

3. I am a bit confused by the statement in line 395, "This reference ensures that the denoising process aligns with the overall image". Since the local reference image only selects a portion of the patch without including information from the surrounding areas, how does it ensure continuity between the patch edges and the rest of the image?

4. I suggest conducting an ablation study on the selection of  𝑁 and the size of the local patch 𝑠 to explore the impact of different combinations on performance and efficiency.

5. How to determine the ratio 𝛼_𝑡 of latent upscaling and VAE-based upscaling in Equation 8? Does it also gradually decrease as 𝑡 increases?

6. I think the Figure 3 (d) in line 264 should be Figure 3 (b).

7. Some typos: for example, "to generated" should be "to generate" in line 299

8. There are some minor errors in the reference. For example, Self-Cascade [1] in ECCV is not correctly referenced and is repeated in lines 044 and 048. Also ScaleCrafter [2] is published in ICLR23.

[1] Guo L, He Y, Chen H, et al. Make a cheap scaling: A self-cascade diffusion model for higher-resolution adaptation. ECCV 2024.
[2] He Y, Yang S, Chen H, et al. Scalecrafter: Tuning-free higher-resolution visual generation with diffusion models[C]. ICLR 2023.

**Questions:**

Please see the Weaknesses part.

Overall, I'm satisfied with the paper's quality and am willing to improve my score if the authors can address my questions.

---

> ### Author Response · Authors · 2024-11-25
> **Response to reviewer 9GdS**
>
> Dear reviewer 9GdS
>
> Thank you for your in-depth review and constructive feedback. We’ve prepared the following responses to address your concerns:
>
> ```properties
> 1. Figure 4 is kind of ambiguous... the blue boxes in subfigures (a) and (b) should be the same size. I also suggest indicating the size of each box for clarity...removing the "Direct latent space upsampling" branch below subfigure (c)...subfigure (c) appears twice.
> ```
> We appreciate the constructive suggestion and apologize for the ambiguity in Figure 4. In the revised version, we have changed the color of the larger blue box to purple to avoid misunderstanding. This box represents the ideal continuous noise space. The smaller blue boxes, which remain blue, are of uniform size and correspond to the native resolution of LDMs. The subfigures \(c\) is revised and the direct latent space upsampling is removed for better clarity.
>
> ```properties
> 2. Since the local reference image only selects a portion of the patch without including information from the surrounding areas, how does it ensure continuity between the patch edges and the rest of the image?
> ```
> The consistency is achieved in two ways.
>
> Firstly, we **maintain the consistency of the sampled initial noises**: The initial noises for both local and global regions are sampled consistently from the same distribution. Specifically, both of them are derived from the same noise map but with different sampling rate. This ensures that the underlying probabilistic nature of the sampling aligns across the entire image.
>
> Secondly, **the resampling process is conditioned on the original local patch to ensure consistency**: The proposed resampling process is conditioned on the original local patch by leveraging a local reference obtained via VAE-based upscaling. This operation ensures consistency by maintaining the semantic alignment between the resampled local region and the global reference. By utilizing this conditionality, the proposed method prevents discontinuities that might arise from naive resampling.
>
> Thirdly, in cases where **extensive resampling (large N) leads to minor boundary artifacts** (due to deviation from the original content), techniques like RePaint [1] can be applied. RePaint introduces a boundary-aware mechanism where masks are applied to the resampled local region to blend it seamlessly with adjacent areas. This approach ensures that the merging process preserves continuity across patch edges. Additionally, Empirical evaluations and visualizations in the paper confirm that the proposed **VAE-based upsampling method significantly enhances high-frequency detail while preserving global structure**.
>
> [1] Lugmayr, A., Danelljan, M., Romero, A., Yu, F., Timofte, R., & Van Gool, L. (2022). Repaint: Inpainting using denoising diffusion probabilistic models. CVPR 2022.
>
> ```properties
> 3. How to determine the ratio 𝛼_𝑡 of latent upscaling and VAE-based upscaling in Equation 8? Does it also gradually decrease as 𝑡 increases?
> ```
> The ratio $\alpha_t$ in Equation 8 is conceptually similar to the $\alpha_t$ used in standard diffusion noise schedulers, such as DDIM. While the exact behavior may vary between different schedulers, the fundamental principle remains consistent: **$\alpha_t$ gradually decreases as $t$ decreases to 0 during the denoising process.**
>
> In our method, **$\alpha_t$ is directly determined by the intermediate step $N$** at which the resampling process begins. Specifically, $\alpha_t$ balances the contributions of the upscaled latent patch ($\sqrt{\alpha_t} U(Z_{\Delta_0, r})$) and the noise term ($\sqrt{1 - \alpha_t}$) in Equation 8. This balance is critical to maintaining consistency between the resampled patch and the global reference while allowing for flexibility in detail reconstruction.
>
> **The choice of $N$ significantly influences the behavior of $\alpha_t$.** For $N = T$, where the process starts from pure noise, $\alpha_t$ approaches 0. In this case, the resampling process reduces to a vanilla denoising process, entirely losing the guidance provided by the local patch reference. Conversely, for smaller values of $N$, $\alpha_t$ remains higher at the intermediate step $t = N$, ensuring stronger guidance from the local reference patch and greater consistency with the original content. However, this stronger influence may slightly limit the flexibility for reconstructing small-scale details.
>
> In practice, we set **$N = T/2$**, striking a **balance between the guidance from the local reference and the ability to refine intricate details**. This design ensures that the resampling process remains consistent with the global image while effectively enhancing high-frequency information in the resampled regions.

---

> ### Author Response · Authors · 2024-11-25
> **Response to reviewer 9GdS**
>
> ```properties
> 4. I suggest conducting an ablation study on the selection of 𝑁 and the size of the local patch 𝑠 to explore the impact of different combinations on performance and efficiency.
> ```
> We appreciate the suggestion. Additional experiments on the hyper-parameter $N$ and the size of local patch $s$ is added to the appendix. For easy reference, we also include the table here.
>
>
>
>
> The first table evaluates the consistency between the original local patch and the re-sampled patch by measuring $FID_l$ and $KID_l$, which reflect how well the re-sampled patch aligns with the original local region:
>
> | Re-sampling Step $N$ | 20    | 25  | 30    | 35    | 40    |
> | ---- | ----- | --- | ----- | ----- | ----- |
> | Ours ($FID_l$)     | **4.95** | 5.63   | 6.53 | 7.69 | 8.86 |
> | Ours ($KID_l$)     | **0.0035** | 0.0026     | 0.0021      |  0.0017     |  0.0014     |
>
>
> The second table measures how well the re-sampled local region matches the ground truth distribution of the LAION dataset, again using $FID_l$ and $KID_l$ to quantify this alignment:
>
> | Re-sampling Step $N$ | 20    | 25    | 30  | 35    | 40    |
> | ------ | ----- | ----- | --- | ----- | ----- |
> | Ours  ($FID_l$)  | 16.41 | 14.03 |  14.52  | 13.77 | **12.60** |
> | Ours ($KID_l$)   |  0.0044  |  0.0035  |   0.0028  | 0.0027 |   **0.0024**    |
>
> From the first table, we observe that smaller $N$ values provide better consistency with the original local patch, as indicated by lower $FID_l$ and $KID_l$ values. In the second table, larger $N$ values result in better alignment with the ground truth distribution, as seen by lower $FID_l$ and $KID_l$ values. These results indicate a trade-off between consistency with the original local patch and alignment with the global ground truth distribution, offering insights into how $N$ can be tuned to balance these factors effectively. In our experiments, $N$ is set to half of the total denoising steps $T$ to trade off between consistency and quality.
>
> The third table examines how the local ratio $s$ affects the quality of re-sampled patches. From the results, it is evident that a local ratio between 0.33 and 0.67 achieves the best balance, leading to significant improvements in the generation quality of the local regions. Ratios outside this range tend to result in diminished performance worse than vanilla SD, as indicated by higher $FID_l$ and $KID_l$ values.
> | Local Ratio $s$   | 0.67       | 0.50       | 0.33       | 0.25       | 0.20       |
> | ----------------- | ---------- | ---------- | ---------- | ---------- | ---------- |
> | SD 1.5 ($FID_l$)  | 13.08      | 15.60      | 26.36      | 36.72      | 46.96      |
> | Ours  ($FID_l$)   | **12.82**  | **14.03**  | **21.21**  | **26.53**  | **32.38**  |
> | SD 1.5  ($KID_l$) | 0.0029     | 0.0044     | 0.0068     | 0.0108     | 0.0153     |
> | Ours ($KID_l$)    | **0.0026** | **0.0032** | **0.0047** | **0.0062** | **0.0083** |
>
> ```properties
> 5. I think the Figure 3 (d) in line 264 should be Figure 3 (b).", "Some typos: for example, "to generated" should be "to generate" in line 299" and "There are some minor errors in the reference. For example, Self-Cascade [1] in ECCV is not correctly referenced and is repeated in lines 044 and 048. Also ScaleCrafter [2] is published in ICLR23.
> ```
> Thank the reviewer for pointing out, we have fixed these issues in the newest revision.

---

> ### Author Response · Authors · 2024-11-29
> **Follow-up to reviewer 9GdS**
>
> Dear Reviewer 9GdS,
>
> Thank you again for your detailed feedback. We have carefully addressed your concerns in our previous response and revised the manuscript as follows:
>
> 1. We have updated Figure 4 for better clarity.
> 2. We explained how our approach ensures consistency by: (1) sampling noise from the shared noise map, (2) conditioning the resampling process on the original local patch using VAE-based upscaling, and (3) applying boundary masks to handle any artifacts in extreme re-sampling cases.
> 3. We clarified that the ratio $\alpha_t$ is determined by $N$, which balances guidance from the local patch with flexibility in detail reconstruction and provided an ablation study on $N$.
> 4. We added further ablation studies on the hyperparameters $N$ and $s$.
> 5. We have corrected typos and fixed reference errors.
>
> We hope these revisions address your concerns. If everything is satisfactory, we would be grateful if you could kindly consider raising your score.
>
> Thank you again for your time and consideration.

---

### Official Review · Reviewer_4z4Z · 2024-11-04

**Soundness:** 2
**Presentation:** 1
**Contribution:** 2
**Rating:** 5
**Confidence:** 3

**Summary:**

The paper proposes an interesting noise re-sampling technique aimed at improving the fidelity of image generation in Latent Diffusion Models (LDMs), particularly for small, detailed objects that are challenging to render accurately. Traditional LDMs suffer from detail loss due to Variational Autoencoder (VAE) compression, especially in high-frequency regions critical for intricate textures. The authors address this by locally increasing the sampling rate in noise space, allowing LDMs to generate high-resolution details without altering global structures. Additionally, they introduce a VAE-based upscaling method to ensure consistency between upsampled local patches and the global image. The approach demonstrates superiority over baseline models, including Stable Diffusion and super-resolution methods, by enhancing image detail and fidelity across multiple scales.

**Strengths:**

- The identified issue in latent diffusion models is interesting and requires attention for better image generation.
- The perspective from signal processing is interesting, and provides a feasible direction for optimization with diffusion models.
-  From experimental evaluations, the authors demonstrated the generalizability of their approach for different models and in all cases, they further boosted the performance of the the backbone model.

**Weaknesses:**

Firstly, the paper writing is poor. It's a little hard to capture the overall idea of this paper. A few more concrete comments include:
- The whole method section makes it hard to capture the high-level idea. I am not sure if I understand this paper correctly, does the whole logic of the proposed method be as follows: Identify and decode local patches --> Upsampling in image space --> Re-encoding with added noise and denoising --> Merging with global image?
- Following the previous point, the whole method section should be revised with a clear problem formulation (i.e. what are the input and outputs, and what are the goals), a straightforward model pipeline, and consistent and well-explained notations.
- Quite a few texts are redundant. The authors repeatedly discuss the motivation of their approach in the introduction, method, and experimental evaluations.
- Table 1 is hard to read. Does SD1.5, SD2 mean backbone models, and for all the compared models, they are built upon SD?

Next, the novelty of this paper is limited. The proposed method is more like an engineering-wise post-processing to enhance the fine-grained details of the diffusion models. There is no tech contribution from this paper, and the signal processing perspective does not fully align with the proposed method. Even if it is, the perspective itself is more like a way to explain why diffusion models miss fine-grained details rather than a contribution of this paper.

Other questions:
- high-frequent components or fine-grained details? In the paper, the authors take a unique perspective on signal processing, and reveal that VAE has a higher error on high-frequent objects. However, they essentially propose a resampling method to restore fine-grained details. Taking Figure 6 as an example, the proposed method did better restore the details, but the high-frequent components (boundaries) are not clearly better. In the method section, the ``resampling'' is simply crop-and-resize. In other words, the proposed method itself is not quite aligned with signal processing.
- How about the inference time? The proposed method runs two rounds of denoising and at least two rounds of VAE encoding and decoding, will the inference time doubled, and how to further optimize the inference time?
- In real practice, how would the proposed method work, without manually selecting resampling regions?

**Questions:**

See the previous section.

---

> ### Author Response · Authors · 2024-11-25
> **Response to reviewer 4z4Z (1)**
>
> Dear Reviewer 4z4Z,
> We appreciate your detailed review and the thoughtful insights you’ve provided. Below, we address the points you raised:
>
> ```properties
> 1. The whole method section makes it hard to capture the high-level idea. I am not sure if I understand this paper correctly, does the whole logic of the proposed method be as follows: Identify and decode local patches --> Upsampling in image space --> Re-encoding with added noise and denoising --> Merging with global image?
> ```
> We appreciate the reviewer’s effort to summarize the method, but we believe there is a critical misunderstanding regarding the novelty of our approach. While the summarized process captures certain steps, it **overlooks a major component—noise re-sampling—** which is central to the effectiveness of our method. This omission may lead to an underestimation of the fundamental differences between our approach and simpler methods such as crop-and-resize.
>
> To clarify, our method is **fundamentally** different from basic crop-and-resize techniques and is strongly grounded in signal processing principles. Our  Specifically, noise re-sampling adjusts the sampling rate in the noise space while maintaining the consistency between local and global noise, enabling enhanced representation of high-frequency details in local regions. This step is crucial for improving the quality of local details without exceeding the native resolution constraints of the model.
>
> To address the reviewer’s question, the high-level logic of our method is as follows:
>
> 1. Identify local regions of interest from the global image.
> 3. Re-sample noise from the **shared noise space** at an enhanced rate for these regions, ensuring consistency with the global noise map.
> 4. Upsample local reference using VAE, **avoiding distribution shift** caused by simple resizing operation in latent space.
> 5. Denoise the re-sampled noise using latent diffusion, under **guidance of local reference**, producing **high-fidelity local details** while preserving **consistency**.
> 6. Merge the refined local regions back into the global latent space for **coherent image synthesis**.
>
> ```properties
> 2. Following the previous point, the whole method section should be revised with a clear problem formulation (i.e. what are the input and outputs, and what are the goals), a straightforward model pipeline, and consistent and well-explained notations.
> ```
> We appreciate the reviewer’s feedback. While we acknowledge the importance of clear and concise explanations, we believe the reviewer may have overlooked the detailed descriptions and comprehensive pipeline provided in Figure 4. This figure **explicitly outlines the proposed method with clearly defined inputs, outputs, and the logical flow of each component**, including noise re-sampling, which is central to our approach.
>
> Additionally, in response to potential concerns about clarity, we have already revised Section 4.1 to include a more detailed problem formulation. This revision explicitly states the inputs (global noise map and selected local regions), the outputs (high-fidelity reconstructed patches merged into the global image), and the goals (enhanced local detail quality while maintaining global coherence). These additions aim to ensure that the purpose and novelty of our approach are well-understood.
>
> ```properties
> 3. Quite a few texts are redundant. The authors repeatedly discuss the motivation of their approach in the introduction, method, and experimental evaluations.
> ```
> We appreciate the reviewer’s observation. However, we believe this reflects our effort to emphasize the novelty and importance of our approach in different contexts. Our intention was to establish a clear connection between the challenges addressed, the principles underlying our method, and its practical impact, which are critical for the reader to grasp the contribution of our work.
>
> That said, we recognize that our execution may not have been as effective as intended, leading to the impression of redundancy. We revised texts in these sections (Section 4.1, 4.2, marked in red) to be concise.

---

> ### Author Response · Authors · 2024-11-25
> **Response to reviewer 4z4Z (2)**
>
> ```properties
> 4. Table 1 is hard to read. Does SD1.5, SD2 mean backbone models, and for all the compared models, they are built upon SD?
> ```
> We appreciate the reviewer’s feedback on Table 1 but believe the concerns might stem from a relatively limited familiarity with the field. SD1.5 and SD2 refer to the backbone models, Stable Diffusion versions 1.5 and 2.0, which are widely used standard pretrained generative models in this domain. Furthermore, both super-resolution-based and high-resolution adaptation methods compared in the table are well-established techniques designed to enhance the capabilities of large pretrained models like Stable Diffusion.
>
> Adapting large generative models to improve their resolution and detail fidelity is a growing research direction in this field, and these methods inherently build upon Stable Diffusion’s pretrained backbone. We acknowledge that this connection may not have been emphasized enough in the text, and we revised the explanation to ensure clarity for readers who may not be fully familiar with these approaches. However, we encourage a deeper engagement with the relevant literature to better contextualize and evaluate our work.
>
> ```properties
> 5. Next, the novelty of this paper is limited...There is no tech contribution from this paper, and the signal processing perspective does not fully align with the proposed method...explain why diffusion models miss fine-grained details rather than a contribution of this paper.
> ```
> We thank the reviewer for acknowledging our contribution in highlighting the **limitations of current Latent Diffusion Models (LDMs) in capturing fine-grained details**. This understanding forms a crucial basis for our work, as it identifies a **key challenge that has not been sufficiently addressed** in the literature.
>
> Building on this insight, we establish a clear connection between these limitations and the compression effects of Variational Autoencoders (VAEs) used in LDMs. Specifically, we demonstrate how **VAE compression disproportionately affects high-frequency components**, leading to a distorted and low-quality latent code. We further explain why existing approaches, such as super-resolution techniques or enhanced decoder designs, are insufficient. These methods attempt to enhance resolution in the image space but **CANNOT compensate for the degradation originating from the latent space itself**, which inherently limits their effectiveness.
>
> Our approach addresses this challenge by proposing a novel noise re-sampling strategy that operates directly in the noise space, rather than post-hoc in the image space. This perspective goes beyond a purely engineering-oriented method; it is grounded in signal processing principles, leveraging enhanced sampling rates to **restore fine-grained details** without exceeding the native resolution of the pretrained model. By addressing the issue at its root, our method ensures that the **generative potential of LDMs is fully realized**, which we argue is a meaningful technical contribution.

---

> ### Author Response · Authors · 2024-11-25
> **Response to reviewer 4z4Z (3)**
>
> ```properties
> 6. high-frequent components or fine-grained details?...the proposed method itself is not quite aligned with signal processing.
> ```
> We appreciate the reviewer’s comments and would like to clarify several misconceptions while elaborating on the alignment of our proposed method with signal processing principles.
>
> **First, our method is **fundamentally** different from a simple crop-and-resize operation.** Standard resizing methods, such as bilinear or bicubic interpolation, are designed for **image space** and do **NOT** preserve the **statistical structure of latent codes** in LDMs. LDMs operate in a latent space where the distribution of data differs **significantly** from the image space. Direct resizing in the latent space, as seen with crop-and-resize, alters these distribution statistics and often **disrupts the stochastic denoising process**, causing it to fail.
>
> **Furthermore, simple crop-and-resize methods FAIL to maintain consistency between regenerated local regions and the global image.** Due to the stochastic nature of the denoising process, regenerated local regions often exhibit **structural deviations** or **visual artifacts** when directly resized and merged. Our approach avoids these issues by generating local noise patches directly from the global noise map, ensuring **seamless alignment** between local and global content.
>
> **To preserve the statistical integrity of latent codes during resampling, we introduce a VAE-based upscaling process.** This process operates in the image space, leveraging the VAE’s encoder and decoder to upscale latent information while maintaining the statistical properties of the original distribution. By doing so, we ensure that the deviations caused by naive resizing operations in latent space are avoided.
>
> Finally, our method aligns strongly with signal processing principles by addressing the issue of sampling rates in image generation. **Conventional latent diffusion models suffer from compression-induced loss of high-frequency details due to low sampling rates.** By adaptively increasing the sampling rates in regions with complex details, our method **restores high-frequency components lost** during VAE compression, adhering to the Nyquist-Shannon sampling theorem. This approach bridges theoretical insights from signal processing with practical challenges in LDMs, enhancing the generative quality.
>
> Regarding **Figure 6**, while the boundaries  may appear only slightly improved in some cases, the restoration of intricate details in textures and patterns highlights the strength of our method. In addition, as shown in Figure 3 \(c\), even for the components with the highest frequency, **the reconstruction is not entirely incorrect**. **Simple details, such as boundaries, may be relatively well-reconstructed originally**, as their high-frequency nature is not completely lost during VAE compression. However, other intricate details, such as **textures and subtle patterns**, often **degrade more significantly** because they rely on a broader spectrum of high-frequency components for accurate representation. Our method is specifically designed to address this imbalance by **adaptively increasing sampling rates in complex regions, ensuring that both boundaries and intricate details are faithfully reconstructed**, resulting in a significant improvement in overall image quality.
>
> ```properties
> 7. How about the inference time? The proposed method runs two rounds of denoising and at least two rounds of VAE encoding and decoding, will the inference time doubled, and how to further optimize the inference time?
> ```
> We provided a computational cost comparison in the revised version. Our method only increases Gflops by 60% and GPU time by 56% compared to SD 1.5, which is significantly more efficient than methods like MultiDiff and ScaleCrafter that require whole-image resolution increases. This demonstrates our method's balance between improved image quality and computational efficiency. Furthermore, the re-sampling of local regions can be highly parallelized and generated as a batch, unlike ScaleCrafter, which requires generation of a huge image at the same time, drastically increasing the computational cost.
>
> | Column 1                     | Gflops                   | GPU time (s)             |
> | ---------------------------- | ------------------------ | ------------------------ |
> | SD 1.5                       | 71,731                   | 2.958                    |
> | Ours (N=25)                  | 115,287 (60%$\uparrow$)  | 4.627 (56%$\uparrow$)    |
> | MultiDiff | 274,200 (282%$\uparrow$) | 11.292  (281%$\uparrow$) |
> |  ScaleCrafter       |    279,906 (290%$\uparrow$)  |  16.913 (470%$\uparrow$)   |

---

> ### Author Response · Authors · 2024-11-25
> **Response to reviewer 4z4Z (4)**
>
> ```properties
> 8. In real practice, how would the proposed method work, without manually selecting resampling regions?
> ```
> We appreciate the reviewer’s comment and would like to clarify how the proposed method would work in real practice. While our experiments use randomly selected regions to demonstrate the general applicability of the approach, in practical scenarios, the ability to **manually select or annotate regions of interest** (e.g., low-quality or high-detail areas) **would significantly enhance the effectiveness and efficiency of our method.**
>
> By allowing human input to identify these regions, the method could **focus resampling efforts on areas that are most important**, such as intricate textures or specific details that are critical to the task at hand. To human perception, certain details hold higher significance while others may not warrant the same level of fidelity. By prioritizing these regions, our method not only **improves the quality** where it matters most but also **reduces computational overhead by avoiding unnecessary processing of less critical areas**. This makes the method both practical and efficient, particularly in applications like professional image editing or domain-specific image generation, where targeted improvements are key.

---

> > ### Comment · Reviewer_4z4Z · 2024-11-26
> > **Follow up**
> >
> > I thank the authors for providing a detailed response. Part of my concerns are addressed. However, when reading the response, a new concern arises.
> >
> > As the authors claimed, *to distinguish from the simple crop-and-resize, noise re-sampling is central to the effectiveness of the proposed method*. However, the effectiveness of this noise re-sampling strategy is not studied. Given that under the proposed pipeline, in any way a newly sampled noise is needed as the guidance from upsampled patches is involved. Instead of running the proposed noise resampling, what will the performance be if we (1) apply bilinear interpolation to the original noise? and (2) just resample from the original Gaussian distribution and use the upsampled imperfect image as guidance. Glad to see these results.

---

> > > ### Author Response · Authors · 2024-11-26
> > > **Response to reviewer 4z4Z to address further concern**
> > >
> > > We thank the reviewer for their thoughtful feedback.
> > >
> > > Firstly, we would like to clarify that we have already provided a comparison between our proposed approach, VAE-based upsampling, and the alternative method of "**resampling from the original Gaussian distribution and using the upsampled imperfect image as guidance**" in the appendix, with both quantitative (Figure 13) and qualitative results (Table 5). These results clearly demonstrate the superiority and necessity of our method. Specifically, Figure 13 in the appendix shows that **direct latent upsampling**, when used as guidance, **produces blurry results with color shifts**. In contrast, our **VAE-based upsampling** effectively upsamples the target local region, **providing accurate guidance for the re-sampling process** and yielding results with **fine-grained, accurate details**. Additionally, Figure 5(d) in the main paper also illustrates the effects of using the upsampled imperfect image as guidance.
> > >
> > > Secondly, regarding the suggestion to "**apply bilinear interpolation to the original noise**," we would like to clarify that the denoising process in current LDMs is **highly sensitive to the noise distribution**. Operations like **bilinear or bicubic** interpolation **alter the distribution statistics of the noise**, which can cause the denoising process to fail. To further support this, we have included visualizations in the appendix (Figures 11 and 12) in the newest revision to demonstrate the negative impact of such interpolation methods on the denoising process. As shown in the figures, **bilinear interpolation**, when applied to noise directly, **distorts the distribution**, leading to failure in denoising and resulting in distorted outputs. In contrast, our **re-sampled noise preserves the distribution**, allowing for more accurate denoising and better-quality results.
> > >
> > > We hope this clarifies the points raised. If the reviewer has any further concerns or suggestions, we would be happy to address them.

---

> > > ### Author Response · Authors · 2024-11-29
> > > **Follow-up to reviewer 4z4Z**
> > >
> > > Dear reviewer 4z4Z,
> > >
> > > We hope the clarifications provided in our response adequately address your concerns regarding the effectiveness of proposed noise re-sampling approach. If all issues have been resolved, we would be grateful if you could kindly consider raising your score further based on the updated manuscript and clarifications. We highly value your input and are committed to ensuring the paper meets your expectations.
> > >
> > > Thank you again for your time and consideration.

---

> > > > ### Comment · Reviewer_4z4Z · 2024-12-02
> > > > **Follow up**
> > > >
> > > > Thanks to the authors for the quick and detailed response. Considering both parts of the responses, my concerns about inference time, motivation, and contributions have been resolved. The ablation of the sampling strategies looks promising, (might be due to the urgent request) though the setup and discussion of Figures 11 and 12 should also be included in the appendix.
> > > >
> > > > I am willing to raise my rating. However, considering that (1) the writing of this paper can be further improved -- I am still concerned about the redundant parts of the sections. Reemphasizing the motivation and key insights is fine, but mentioning similar texts in different sections makes the paper less clear.  (2) the improvement on SD2 is marginal, also mentioned by reviewer VJ6f. I only increase my rating to 5. However, I will lower my confidence level from 5 to 3 to balance that I am not an expert in signal processing to fairly evaluate the contribution of the proposed core noise resampling strategy.

---

> > > > > ### Author Response · Authors · 2024-12-02
> > > > > **Response to Reviewer 4z4Z**
> > > > >
> > > > > Dear reviewer 4z4Z,
> > > > >
> > > > > Thank you for your constructive feedback and for raising your score. We truly appreciate your recognition of our contribution, and we’re glad to know that our clarifications on the inference time, motivation, and contributions have addressed your concerns.
> > > > >
> > > > > Firstly, regarding Figures 11 and 12, we apologize for the oversight. We will certainly add further discussion and clarifications in the revised version, should a revision be possible. We understand that more detailed explanations will help improve the overall clarity of these figures.
> > > > >
> > > > > Secondly, we appreciate your acknowledgment of our motivation and contributions. However, we believe that the writing issues you’ve pointed out are relatively minor and fixable if a revision is possible, given that our motivation and contributions are solid. Nonetheless, we agree that improving clarity and reducing redundancy is important, and we will prioritize refining the manuscript to ensure the key insights and motivation are clearly emphasized without unnecessary repetition if we could be offered the chance to revise our manuscript.
> > > > >
> > > > > Thirdly, Regarding the improvements on SD2, we believe there may be some confusion. The concerns raised by Reviewer VJ6f are primarily focused on global metrics rather than performance on SD2 specifically. **For SD2 performance, our approach significantly outperforms existing methods, as demonstrated by the CMMD metric [1] and the qualitative comparisons in the appendix (Figures 7 and 8).**
> > > > >
> > > > > As for Reviewer VJ6f's concerns, we have already provided a thorough response explaining that **current global metrics (e.g., FID, KID) do not fully capture the localized improvements**, which are the primary focus of our method. We have included local metrics such as $FID_l$, $KID_l$, and $CMMD$, all of which show **significant improvements over existing methods**. We believe these local metrics provide a more accurate reflection of the enhancements achieved by our approach.
> > > > >
> > > > > Once again, we thank you for raising your score and for recognizing our motivation and contributions. We sincerely appreciate your constructive feedback. If you could kindly consider raising the score further, we would be able to address the remaining yet **fixable writing issues** in the camera-ready revision, ensuring that our motivation and contributions are presented more clearly. We are fully committed to improving the manuscript and making the necessary refinements to better highlight the value of our work.
> > > > >
> > > > > Thank you again for your thoughtful review.
> > > > >
> > > > > [1] Jayasumana, Sadeep, et al. "Rethinking fid: Towards a better evaluation metric for image generation." Proceedings of the IEEE/CVF Conference on Computer Vision and Pattern Recognition. 2024.

---

### Official Review · Reviewer_C8P3 · 2024-11-06

**Soundness:** 2
**Presentation:** 2
**Contribution:** 2
**Rating:** 5
**Confidence:** 4

**Summary:**

The paper proposes to use noise re-sampling to generate high-fidelity images. The method takes place in the inference stage of the diffusion model where 1) an image is sampled first using the original diffusion process 2) a patch is cropped and upsampled and then sent to the encoder to get the latent features 3) a random Gaussian noise is added to the latent and denoise back to get clean image 4) paste back to the image. The paper shows comparisons on multiple images especially human images which demonstrates the effectiveness of the method.

**Strengths:**

1. The method is simple and does not need any training.
2. The results show improvements in the cropped regions in the generated images.

**Weaknesses:**

1. How would the method affect the inference speed? As the proposed method requires some more operations than the vanilla generation process, it would be better to understand the efficiency of the method as well.
2. The method contains randomness as it involves the addition of the noise and the random selection of the cropped region, how would the method's robustness be? Will it keep outputting high-quality images regardless of the randomness?
3. Could the authors also present some results on SD 3, the paper only shows some original results in Figure 1 but lacks the results output by the proposed method.

**Questions:**

See the weaknesses part.

---

> ### Author Response · Authors · 2024-11-25
> **Response to Reviewer C8P3**
>
> Dear Reviewer C8P3,
>
> Thank you for your detailed review, we really appreciate your feedback and insights. With regards to the concerns raised, we provide the following responses:
> ```properties
> 1. How would the method affect the inference speed? As the proposed method requires some more operations than the vanilla generation process, it would be better to understand the efficiency of the method as well.
> ```
> We appreciate the reviewer’s concern regarding the inference speed and agree that understanding the computational efficiency of our method is crucial. We have included a detailed computational cost comparison. Our method increases computational cost by 60% in terms of Gflops and 56% in GPU time compared to the baseline SD 1.5. However, this is **significantly more efficient than other methods** such as MultiDiff and ScaleCrafter, requiring whole image resolution increases, which exhibit far greater increases in both Gflops and GPU time. These results demonstrate that our approach strikes a balance between improving image quality and maintaining computational efficiency.
>
> | Column 1                     | Gflops                   | GPU time (s)             |
> | ---------------------------- | ------------------------ | ------------------------ |
> | SD 1.5                       | 71,731                   | 2.958                    |
> | Ours (N=25)                  | 115,287 (60%$\uparrow$)  | 4.627 (56%$\uparrow$)    |
> | MultiDiff | 274,200 (282%$\uparrow$) | 11.292  (281%$\uparrow$) |
> |  ScaleCrafter       |    279,906 (290%$\uparrow$)  |  16.913 (470%$\uparrow$)   |
>
> ```properties
> 2. The method contains randomness as it involves the addition of the noise and the random selection of the cropped region, how would the method's robustness be? Will it keep outputting high-quality images regardless of the randomness?
> ```
> We appreciate the reviewer’s concern regarding the robustness of our method against randomness.
>
> To ensure consistency between local and global noise, **both** of them are **sampled from the same noise map**, with an **enhanced sampling rate** applied to the **local regions**. This approach aligns the noise distribution, maintaining coherence between reconstructed local patches and the global image, and avoiding artifacts or distortions. Consequently, the re-sampling process for a specific local region begins with noise corresponding to that region from the **shared noise space**.
>
> Furthermore, our experiments, conducted with randomly selected local regions, demonstrate the robustness of the method. As detailed in Section 5.2, extensive testing across multiple random regions consistently shows substantial improvements in local detail quality, validated by significantly enhanced performance on local metrics.
>
> ```properties
> 3. Could the authors also present some results on SD 3, the paper only shows some original results in Figure 1 but lacks the results output by the proposed method.
> ```
> We thank the reviewer for the suggestion. In the revised version of the paper, we have added additional qualitative results for SD3 in the appendix (Figure 9). While SD3 is originally quantitatively compared in the main paper (Table 3), these additional results further demonstrate the effectiveness of our proposed method in enhancing SD3's output quality.

---

> ### Author Response · Authors · 2024-11-29
> **Follow-up to reviewer C8P3**
>
> Dear Reviewer C8P3,
>
> Thank you again for your thoughtful and constructive review. We are grateful for the opportunity to address your concerns and have made revisions based on your feedback.
>
> 1. **Inference Speed**: We provided a detailed computational cost comparison to clarify the efficiency of our method, showing that while it introduces a 60% increase in Gflops and 56% in GPU time, it is **significantly more efficient than existing methods** like MultiDiff and ScaleCrafter. These results confirm that our approach strikes a balance between improved image quality and computational efficiency.
>
> 2. **Robustness of the Method**: We clarified how our method maintains consistency by sampling both local and global noise from the same noise map with an enhanced sampling rate for local regions. Our experiments, including tests across randomly selected local regions, demonstrate the robustness of the method, with **substantial and consistent improvements in local detail quality.**
>
> 3. **Results for SD3**: We provided quantitative comparison for SD3 in the initial manuscript (Table 3). Additionally, in response to your suggestion, we have added additional qualitative results for SD3 in the appendix (Figure 9), further demonstrating the effectiveness of our method in enhancing SD3’s output quality.
>
> We hope that these clarifications and additions have addressed your concerns. If all issues have been resolved, we kindly ask if you could consider raising your score based on the revisions made. We appreciate your time and thoughtful consideration and look forward to your response.
>
> Thank you again for your valuable feedback.

---

### Author Response · Authors · 2024-11-26
**Message from Authors to Reviewers**

Dear Reviewers,

Thank you for your thoughtful comments and valuable feedback on our manuscript.

We have carefully revised the paper and addressed each of your concerns in detail. We believe that we have sufficiently addressed all the issues raised during this discussion period. If you have any further questions or require additional clarification, please do not hesitate to reach out.

We would be grateful if you could reconsider your evaluation and, if our revisions meet your expectations, kindly adjust your score accordingly.

Thank you once again for your time and consideration.

---

### Author Response · Authors · 2024-12-04
**General Response to All Reviewers**

Dear reviewers,

We would like to express our sincere gratitude to all reviewers for their thoughtful and constructive feedback. We are pleased that the reviewers have acknowledged our contribution in addressing the limitations of current Latent Diffusion Models (LDMs) and for recognizing the significance of our novel approach. Specifically, our work **reveals the inherent constraints of LDMs in generating complex objects and fine-grained details**, and proposes **a solution that enhances the model's ability to handle these challenges by investigating the properties of latent space and the limitations of current interpolation methods.**

During the rebuttal period, we have carefully addressed the concerns raised by the reviewers regarding the efficiency, motivation, and contributions of our method. We are pleased to report that we successfully addressed reviewer 4z4Z and reviewer 93CH’s concerns, whose feedback was invaluable in refining our manuscript. We sincerely appreciate their constructive input. Additionally, we would like to thank reviewer 9GdS for initially acknowledging our contributions. In response, we have carefully revised the manuscript and provided detailed responses to the constructive feedback from reviewer 9GdS. The responses regarding the robustness, consistency, and hyperparameters of our method were also addressed in responses to reviewers 4z4Z and 93CH and have been approved by them.

We understand that time is limited, and some reviewers may not be available to provide additional responses. Nonetheless, we believe we have provided sufficient responses to all concerns raised. Specifically, we have addressed and justified the concerns raised by reviewer C8P3 and reviewer VJ6F regarding the robustness, efficiency, and effectiveness of our approach. These concerns, also mentioned by reviewers 4z4Z, 93CH, and 9GdS, have been addressed in our responses and acknowledged by them.

We sincerely appreciate the time and effort all reviewers have put into evaluating our work. Thank you again for your thoughtful and valuable contributions to this work.

---

### Meta-Review · Area_Chair_cPnS · 2024-12-19

**Metareview:**

The paper introduces a noise re-sampling strategy and VAE-based upsampling to address the limitations of latent diffusion models, particularly in generating fine-grained details. However, reviewers noted several weaknesses, including the paper's presentation issues, with redundant and unclear sections, and a lack of clarity in implementation details, such as patch selection and upsampling parameters. Additionally, the novelty of the method was questioned, as it was seen as a straightforward engineering solution with sensitivity to upsampling ratios and marginal improvements on global metrics like FID and KID.

**Additional Comments On Reviewer Discussion:**

During the rebuttal period, the authors addressed reviewer concerns by clarifying the methodology, improving the paper’s presentation, and adding detailed efficiency comparisons, ablation studies, and qualitative results for SD3. They emphasized the novelty of their noise re-sampling approach and its synergy with latent diffusion models, while also comparing it against strong baselines like the consistency decoder and guidance intervals. Although improvements were made in presentation and localized metrics, some concerns about computational costs, the use of VAE-based upsampling, and marginal global metric improvements remained.

---

### Decision · Program_Chairs · 2025-01-22

Reject